# Unlocking global carbon reduction potential by embracing low-carbon lifestyles

Yuru Guan [1,2], Yuli Shan [2,3] ✉, Ye Hang[2], Qingyun Nie[4], Yu Liu [5,6] & Klaus Hubacek [1] ✉

Low-carbon lifestyles provide demand-side solutions to meet global climate targets, yet the global carbon-saving potential of consumer-led abatement actions remains insufficiently researched. Here, we quantify the greenhouse gas emissions reduction potential of 21 low-carbon expenditures using a global multi-regional input-output model linked with detailed household expenditure data. Targeting households exceeding the global per-capita average required to stay below 2 degrees, our model captures changes in direct energy use, household consumption and upstream intermediate industrial inputs. We find that implementing a combination of low-carbon expenditures among the top 23.7% emitters reduces global carbon footprints by 10.4 gigatons $CO_2$e (i.e., 40.1% of the household consumption-based emissions of the 116 countries analysed in this study or 31.7% of the global total in 2017). Consumption pattern changes related to mobility and services contribute 11.8% and 10.2% of emission reductions. North America shows substantial reduction potential, while some Sub-Saharan African countries present unexpected mitigation possibilities. However, a rebound effect from re-spending income savings from lifestyle changes offsets the expected carbon savings by 6.5% to 45.8% (0.7–4.8 gigatons $CO_2$e).

Recent work has highlighted the growing importance of demand-side mitigation solutions to achieve global climate targets, as supply-side measures cannot be solely relied upon[1–3]. From a consumption perspective, accounting for all upstream emissions along global supply chains, household consumption triggers directly and indirectly around two-thirds of total greenhouse gas (GHG) emissions, thus transitioning household consumption patterns towards low-carbon modes should be a critical part of mitigating climate change[4,5].

The literature on carbon inequality emphasizes the importance of equitable demand-side measures[6]. Addressing climate change requires a multifaceted approach that includes cutting emissions from high emitters while supporting those who face barriers to low-carbon transitions, such as energy poverty. Research found that the top 10% of

emitters accounted for 48% of the global emissions in 2019, with the top 1% contributing 16.9% to the global total, while the bottom 50% emitted 12%[7]. Between 1990 and 2019, the bottom 50% contributed only 16% of global emissions growth, whereas the top 1% accounted for 23% of the total[7]. This disparity highlights that a relatively small, wealthy part of the global population predominantly drives consumption-based emissions[8,9]. This situation underscores the urgent need to propose demand-side measures that specifically target carbon-intensive activities among top emitters, as those households have contributed most to climate change and have the greatest capacity for reducing emissions. A recent study by Büchs et al. exemplifies this by investigating the carbon savings by hypothetically reducing the energy consumption of the top 20% of consumers across

[1]Integrated Research on Energy, Environment and Society (IREES), Energy and Sustainability Research Institute Groningen (ESRIG), University of Groningen, Groningen, the Netherlands. [2]School of Geography, Earth and Environmental Sciences, University of Birmingham, Birmingham, UK. [3]Birmingham Institute for Sustainability and Climate Action, University of Birmingham, Birmingham, UK. [4]School of Management, Nanjing University of Posts and Telecommunications, Nanjing, China. [5]College of Urban and Environmental Sciences, Peking University, Beijing, China. [6]Institute of Carbon Neutrality, Peking University, Beijing, China. ✉e-mail: y.shan@bham.ac.uk; k.hubacek@rug.nl

27 European countries to the level of the 80th percentile. Such an intervention could reduce GHG emissions by 9.7% of these countries' total emissions[10]. Additionally, emerging and developing economies (as classified by the International Monetary Fund[11]), particularly those with large populations and rapidly industrialising sectors, have become significant contributors to global carbon emissions. Within these economies, households with high consumption levels are playing an increasingly prominent role in driving carbon emissions. However, there is limited research focusing on demand-side reduction measures aimed at these high-emitting segments of the population across countries worldwide.

To effectively achieve demand-side mitigation, adopting low-carbon lifestyles that minimise GHG emissions is essential[12,13]. Lifestyle is a multifaceted construct including behaviours, cognitions, and contextual factors[14,15]. Household expenditure often serves as a reliable proxy for individual lifestyles, reflecting choices across transportation, food, housing, and consumer goods[2,16]. By examining consumption-related behaviours, such as product selection and usage patterns, we can better understand the drivers of household carbon emissions. For instance, decisions to purchase durable appliances, consume natural fibre clothing or reduce food waste directly impact household carbon footprint. A deeper understanding of these behaviours is crucial for designing effective interventions to promote low-carbon lifestyles.

The 'avoid-shift-improve' framework provides a holistic perspective on effective low-carbon lifestyle changes that correspond to actions through three distinct approaches: absolute reduction, consumption pattern shift, and efficiency improvement[1,12]. For example, minimising food waste aligns with the 'avoid' approach, focusing on the absolute reduction of food consumption. Shifting from private vehicles to public transportation represents a 'shift' approach, and opting for seasonal food consumption thereby reducing required energy inputs in the agricultural sector can be categorised as an 'improve' approach for upstream industries. These examples illustrate how the 'avoid-shift-improve' framework can guide us in making informed choices and taking actions that contribute to a more sustainable, low-carbon future.

Following this framework, a growing body of literature explores low-carbon lifestyle transitions. Their emphasis primarily centres on a few key domains, including food[17,18], mobility[19,20], and buildings[21–23]. However, many of these assessments have a relatively narrow scope, which focuses only on one or a few of the major domains[24]. A limited number of studies have considered multiple measures, but even these studies examined different measures separately[5,16]. Moreover, many existing studies have centred on high-income countries (as classified by the World Bank[25]) such as European countries[5,17], the United Kingdom[26], the United States[15], and Japan[27], or selected emerging and developing economies such as India and China[28,29]. This preference is mainly driven by richer data availability, higher environmental awareness, and stricter environmental policies in these regions[5,15,17,26,27]. While the measures proposed in these studies, applied uniformly to all populations of these countries, can influence the consumption of carbon-intensive products, they may disproportionately affect vulnerable populations, particularly those already struggling to achieve decent living standards[30,31]. Furthermore, given that top global emitters are not confined to high-income countries but come from all world regions[7], it becomes increasingly important to explore the mitigation possibilities of high-carbon households in emerging and developing economies[5].

When discussing lifestyle changes through consumption-related behaviours, it is crucial to acknowledge that carbon savings are frequently counteracted by rebound effects[32–34]. An example is that residents adopting home insulation measures may result in direct rebounds, such as turning up the thermostat to increase comfort, and/or indirect rebounds, that is spending the remaining savings on other products and services[35]. The direct rebound effect, primarily focusing on direct energy consumption[36], has been extensively studied and factored in the design of energy-saving guidelines or policies, exemplified by initiatives in the United Kingdom and Ireland[32,35]. These guidelines mainly recommend considering additional energy use (e.g., a 20% increase in heating demand[35]) when estimating potential energy savings from interventions like the installation of energy-efficient household boilers. However, there exists a notable gap in comprehending and quantifying broader, indirect rebounds encompassing all upstream processes, mainly due to uncertainties in re-spending patterns and limitations in detailed data on household expenditures[33,37,38].

In this paper, we conduct an assessment of the household carbon footprints across various consumption levels, using a modified household expenditure database derived from the World Bank Global Consumption Dataset (WBGCD) and a global multi-regional input-output dataset from the Global Trade Analysis Project (GTAP)[8,31]. Our analysis covers both direct GHG emissions from home fuel use and private transport and indirect or upstream GHG emissions resulting from household consumption activities creating emissions along global supply chains. This allows us to pinpoint carbon footprint hotspots among population groups, regions, and consumption categories. Households that exceed the global average carbon targets (aiming to stay below 2 degrees) are modelled in low-carbon lifestyle changes. We select 21 low-carbon expenditures across food, diet, mobility, buildings, clothing, manufactured products, and services. We simulate the carbon reduction potential of these 21 lifestyle changes on household direct energy use and other final consumption and upstream emissions along the entire global supply chain[39]. Our analysis quantifies the aggregated carbon reduction potentials achieved through a combination of low-carbon expenditures for 116 countries and looks closer into household-specific mitigation outcomes. Furthermore, we discuss how much of the expected mitigation benefits attributed to these lifestyle changes may be offset by indirect rebound effects under three re-spending scenarios. We aim to enhance the understanding of rebound effects throughout the entire supply chains and shed light on the magnitude of these effects utilising our detailed household expenditure data.

## Results
### Hotspots of global household carbon footprints
We calculated the carbon footprint in 2017 for households with varying consumption patterns of 201 expenditure groups across 116 countries representing 79.5% of global GDP and 87.3% of the global population including 86 emerging and developing economies. This calculation encompasses direct GHG emissions from fuel use in home and private transport and indirect GHG emissions associated with household consumption activities, emitted in production processes along global supply chains.

Figure 1a shows that higher expenditures translate into higher carbon footprints among household deciles on a global scale[7,9]. The carbon footprint of the poorest decile amounts to 0.5 t $CO_2$e per capita in 2017, while the wealthiest decile has an average carbon footprint of 15.6 t $CO_2$e per capita. The size of the household carbon footprint is closely linked with their consumption patterns. In general, expenditure on buildings and food tends to be the largest contributors to GHG emissions. Poorer households predominantly contribute GHG emissions through food consumption, while wealthier households exhibit greater shares of emissions from services and mobility. The distribution of carbon footprints among the four gases varies significantly: food consumption primarily drives $CH_4$ and $N_2O$ emissions, while emissions from F-gases are predominantly linked to the consumption of manufactured products (Fig. 1b).

We derived a range of targets for global annual consumption-based carbon emissions informed by 2-degree-consistent pathways projected by integrated assessment model scenarios[40]. 23.7% of the global population (1.6 billion) have a per-capita carbon footprint

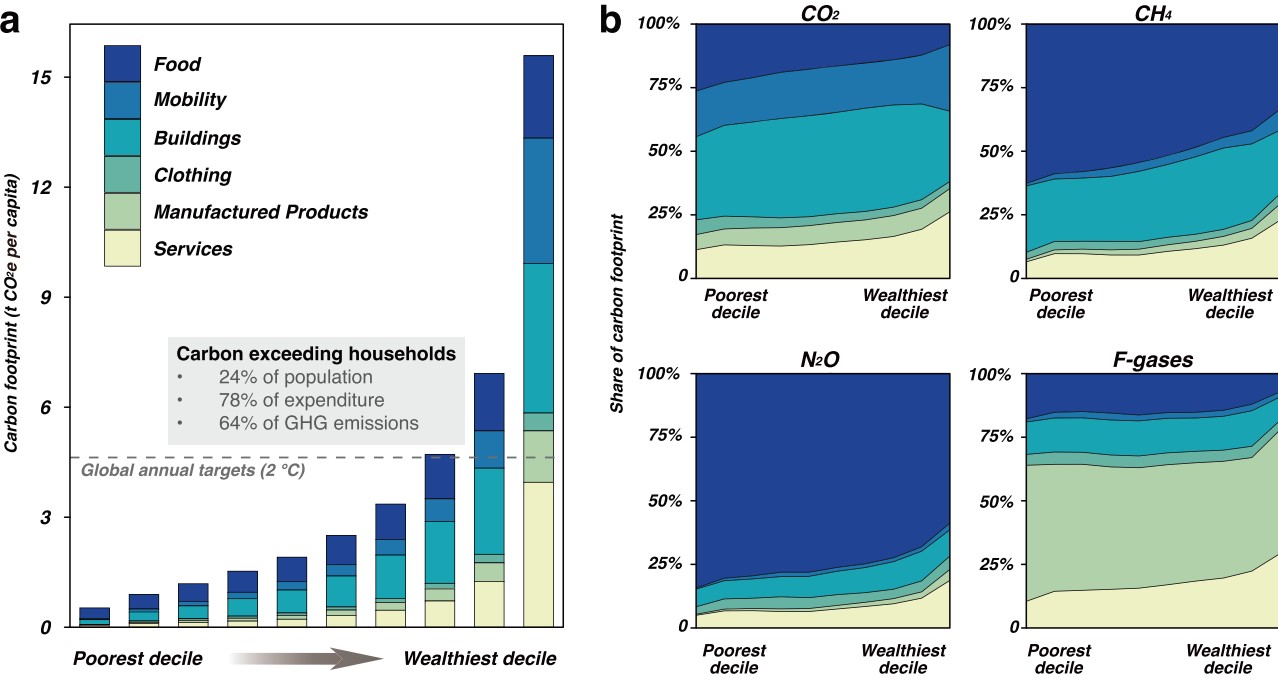

**Fig. 1 | Global household carbon footprints per population decile in 2017. a** Per capita carbon footprint by consumption categories (stacked bars). Mobility covers direct emissions from private transport and indirect emissions induced by public transport. Buildings encompass direct emissions from residential fossil fuel use and indirect emissions from home electricity and energy services use, construction, and building material use. The other four categories cover indirect emissions from all corresponding upstream emissions (see details in Supplementary Data 1). The added dashed line shows a global annual target of 4.6 t $CO_2$e per capita per year for household carbon footprints in 2020, aimed at limiting climate warming to below 2 degrees. **b** Household carbon footprint distribution by categories and gases ($CO_2$, $CH_4$, $N_2O$, and F-gases).

exceeding the global annual average target (i.e., the 2020 upper limit of 4.6 t $CO_2$e per capita), 89.0% of whom live in high- and upper-middle-income countries. The carbon-exceeding households are responsible for about 78.1% of global expenditure and contribute 63.7% of consumption-based emissions of the 116 countries analysed in this study.

Figure 2 shows disparities in carbon footprints across regions. The proportion of carbon-exceeding households varies considerably across regions. North America has the highest average footprint (17.2 t $CO_2$e per capita), with 85.4% of its population exceeding the global 2020 target. Lower-footprint regions such as Sub-Saharan Africa show substantial variation, with 5.4% of the population having a much larger footprint (9.7 t $CO_2$e per capita) compared to its carbon-compliant households. Turning our attention to individual countries, India, the third-largest global consumption-based GHG emitter in 2017 (2.2 gigatons (Gt) $CO_2$e), has only 3.7% of its population comprising carbon-exceeding households. In contrast, Luxembourg, a high-income country with low total emissions (14.2 Mt $CO_2$e), have the largest share of carbon-exceeding households (99.7% of its population). China exhibits both high carbon footprints (5.2 Gt $CO_2$e) and a substantial share of carbon-exceeding households (24.0%). This highlights the complex relationship between regional development, income levels, and carbon footprints. The occurrence of carbon exceedance extends beyond just high-emitting or advanced economies, highlighting the necessity for nuanced strategies that aim to reduce household carbon footprints globally, with a focus on demand-side mitigation measures.

### Reduction potential from engaging in various low-carbon expenditures

Here, we present the results of implementing lifestyle-oriented mitigation measures for carbon-exceeding households across 116 countries. We modelled the reduction potential of 21 low-carbon expenditures separately (see detailed descriptions in Supplementary Data 2). These expenditure-related mitigation measures align with the 'avoid-shift-improve' framework[1,12], addressing carbon reduction through absolute reduction, consumption shift and efficiency improvement. Our results provide a deeper understanding and enable the comparison of the mitigation benefits associated with the adoption of low-carbon expenditures for specific household groups.

Figure 3 shows considerable emission reduction outcomes from the implementation of these lifestyle changes for carbon-exceeding households. The 21 selected low-carbon expenditures offer global GHG reduction potentials ranging from −0.01% ('Natural Materials') to 10.9% ('Nonmarket Services') including avoided upstream emissions. Given that higher costs on leisure activities among carbon-exceeding households, low use of commercial services (abbreviated as 'Nonmarket Services') presents a mitigate potential of 10.9%. In the diet category, shifting towards a healthy vegan diet—reducing consumption of animal-based food, sugar, and unhealthy processed food products—presents a promising reduction potential to reduce global GHG emissions by 8.3%. The four diet-related expenditures focus on reducing specific food consumption (primarily animal-based food) and substituting them with plant-based alternatives, which overlap in their approach but differ in the targeted products (see more in Supplementary Data 2). In the buildings category, implementing passive house standards could result in a 6.0% carbon reduction. Turning to manufactured products, implementing sharing and repair initiatives for home appliances ('Share & Repair') could contribute to a 3.0% reduction. Regarding mobility, adopting modes such as transitioning from private vehicles to public transportation ('Less Cars'), working from home, and halving air travel could potentially reduce carbon emissions by 1.4−3.6%. It is worth noting that working from home reduces emissions from land transport but leads to an increase in home energy use, thus weakening the overall mitigation. In the food category, reducing food waste yields more modest results (1.3%), while opting for seasonal and organic food has minimal impacts (0.1−0.8%).

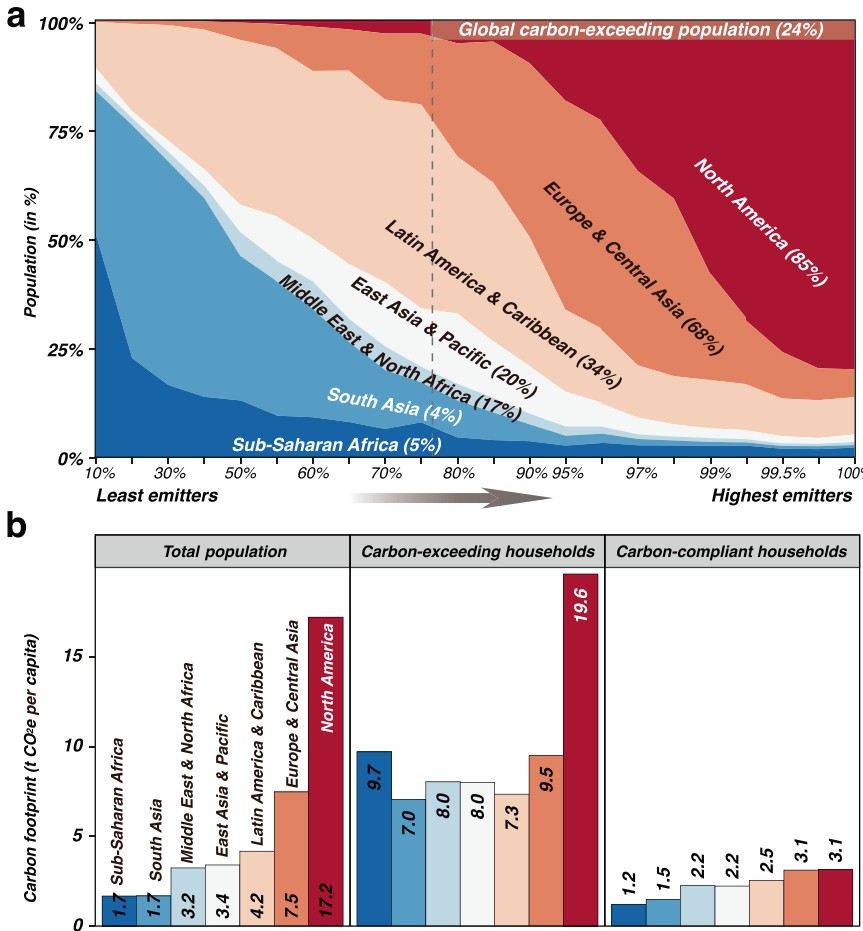

**Fig. 2 | Regional carbon emission distribution in 2017. a** Regional population distribution within global emitter groups, with the carbon-exceeding population shares indicated. **b** Region-average carbon footprints for household groups (i.e., total population, carbon-exceeding households, and carbon-compliant households).

In clothing, changing to fibres has limited impact but extending the lifespan of garments through practices such as swapping and repairing ('Durable Fashion') could achieve a 1.2% reduction in global GHG emissions. Overall, 'Avoid' measures show higher mitigation potential, followed by 'Shift' strategies with moderate impact. 'Improve' measures contribute to reductions but largely rely on production-based advancements for deeper decarbonisation.

We also found that mitigation potentials differ among regions (Fig. 4). The effectiveness of lifestyle changes arises from differences in infrastructural conditions, supply chains with respective production patterns and energy mix, as well as consumption patterns among households and countries. North America and Europe and Central Asia consistently exhibit higher relative reduction potentials across various low-carbon expenditures compared to the global average, whereas South Asia demonstrates the lowest mitigation potential in most changes.

Adopting a healthy vegan diet offers considerable potential for carbon reduction across regions. For example, in Latin America & Caribbean, such a diet could achieve a 17.4% reduction in carbon emissions. This substantial potential aligns with the region's existing dietary challenges, characterised by unhealthy dietary patterns and reliance on carbon-intensive food consumption patterns[18]. Similarly, at the country level, Mongolia stands out with notable reduction potential by adopting different diets. Households in Mongolia contribute considerably to emissions from food consumption, accounting for 59.8% of its total carbon footprint in 2017. When comparing three low-carbon lifestyles related to mobility in each country, 63 out of 116 countries have greater reduction potentials by

adopting 'Less Cars' compared to the other two mobility-related expenditures. 33 countries benefit more from 'Working from Home' and 20 countries see more advantages from 'Less Flying'. Implementing passive house standards can significantly cut residential energy use, leading to considerable carbon reductions across many countries, particularly in North America, and Europe and Central Asia. In some nations, such as Kyrgyzstan and Sweden, 'Durable Fashion' exhibits a relatively higher potential for carbon emissions reduction, with reductions of 3.6% and 2.7%, respectively. Furthermore, embracing 'Share & Repair' practices for home appliances could result in considerable emissions reductions in 94 out of 116 countries, surpassing the potential reductions achieved by adopting 'No Chemicals' and 'Durable Appliances'. Households in North America and Europe & Central Asia can achieve greater carbon reductions compared to other regions through three services-related expenditures: lower use of commercial services, decreased long-distance leisure travel, and proximity-based services.

**Mitigation potentials from combining low-carbon expenditures**
When assessing the cumulative effects of implementing these actions simultaneously, we assumed widespread adoption of all low-carbon expenditures by global carbon-exceeding households. To avoid double counting, we accounted for overlapping impacts, particularly where multiple actions target similar household activities. For example, we selected the diet with the highest reduction potential to avoid overestimating carbon savings due to the overlap among the four diet-related expenditures. Additionally, we excluded actions unlikely to reduce emissions, such as using natural building materials in certain

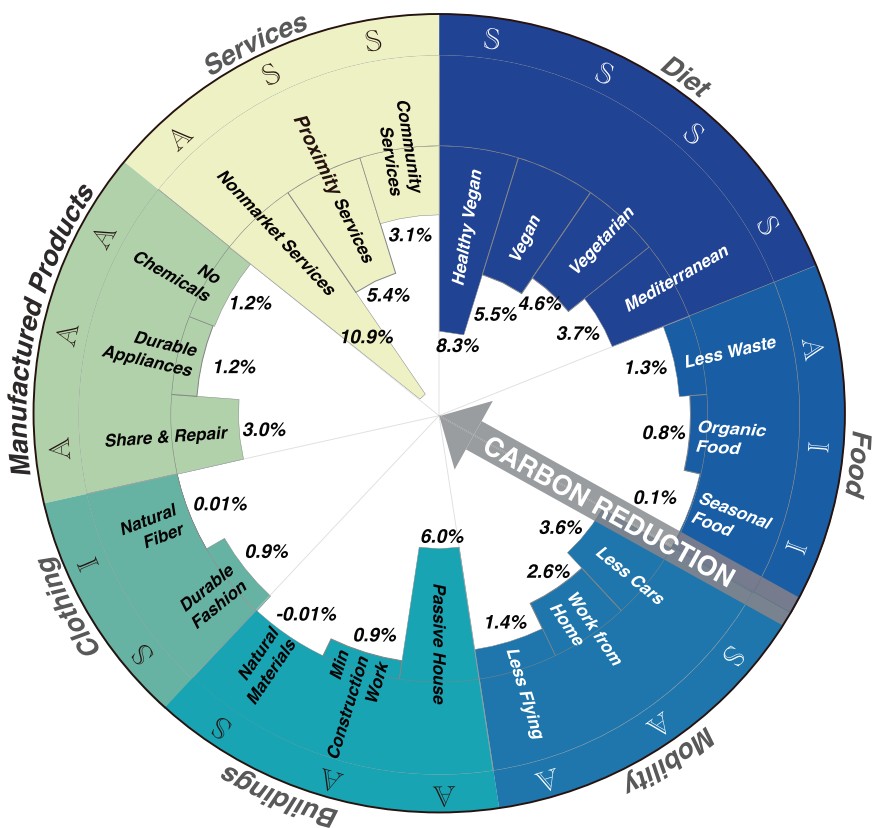

**Fig. 3 | Global carbon reduction potentials of 21 low-carbon expenditures.** Letters 'A', 'S', and 'I' refer to three low-carbon approaches: avoid, shift, and improve, respectively. Overlaps between different expenditures and potential rebound effects are not considered in this figure.

countries. Our aim is to estimate the maximum achievable carbon savings rather than to provide precise predictions.

Globally, implementing a combination of expenditure-focused mitigation measures could potentially lead to a 10.4 Gt $CO_2$e reduction in the carbon footprint, representing 40.1% of the household consumption-based emissions of the 116 countries analysed in this study or 31.7% of the global household carbon footprint in 2017 (Fig. 5). The reduction primarily originates from changes in household consumption volumes and patterns among carbon-exceeding populations, particularly in mobility (3.0 Gt $CO_2$e or 11.8%), services (2.6 Gt $CO_2$e or 10.2%) and food (2.1 Gt $CO_2$e or 8.2%). Changes in clothing expenditures contribute the least (0.2 Gt $CO_2$e or 0.9%). Breaking down by gases, $CO_2$ accounts for the largest share at 29.8%, followed by $CH_4$ (7.9%), $N_2O$ (1.8%), and F-gases (0.6%).

The relative aggregated GHG reduction varies across 116 countries, ranging from 2.3% in Democratic Republic of the Congo to 72.3% in Malta. We observe large relative mitigation potentials for countries in North America (3.7 Gt $CO_2$e or 66.7%), Europe & Central Asia (3.3 Gt $CO_2$e or 53.8%), and Latin America & Caribbean (0.8 Gt $CO_2$e or 41.0%). The United States could achieve the largest absolute reduction (3.7 Gt $CO_2$e) mainly through expenditure changes in mobility and services (see details in Supplementary Fig. 2). For European countries, a set of expenditure-focussed mitigation measures could result in substantial decreases in the carbon footprint, particularly in Luxembourg (67.2%), Denmark (64.6%), and Greece (64.5%). Luxembourg shows huge potential, with reductions of 9.5 Mt $CO_2$e compared to its GHG emissions in 2017, primarily due to the extensive engagement of a substantial portion of its population (99.7% are carbon-exceeding households) in such demand-side mitigation.

Figure 6 illustrates significant deviations (ranging from −7% to 33%) between carbon savings achieved by targeting carbon-exceeding households with a combination of low-carbon expenditure measures

and those obtained by applying the same changes to a corresponding proportion of "average consumers" in each country—a common practice in the literature[41,42]. These deviations highlight the heterogeneity in household consumption patterns, with high-emitting households demonstrating a higher potential for reducing their carbon footprint. In North American and European nations, the observed deviations remain relatively modest despite a high potential for emission reductions. This can be primarily attributed to the substantial presence of carbon-exceeding households within the randomly selected average consumers in these affluent nations. Notably, we found that some Sub-Saharan African countries, such as Mauritius (achieving a 52.5% reduction in this study), Namibia (45.6%), and Chad (44.7%), often overlooked in previous studies, display relatively higher deviations. Namibia's reductions are driven by changes in food, mobility and services expenditures, due to its heavy dependence on the tourism industry.

Figure 7 shows different relative reduction potentials across population groups with varying expenditure levels. For instance, households in China display a median reduction of 52.13% (with the 25th−75th percentile a reduction ranging from 52.07% to 52.14%), while in Angola, this median stands at 54.5% (with the 25th−75th percentile between 44.0% and 69.8%). We also observe a regressive distribution pattern in several countries where lower household expenditure levels correlate with higher relative reduction potential in countries such as Angola, South Africa, Luxembourg, and Eswatini. This pattern does not universally apply, as seen in other nations such as the United States, Finland, Kazakhstan, and China, where wealthier households may benefit more from lifestyle-oriented measures in terms of carbon reduction than others. Additionally, certain countries show unique distributions; for instance, in Belgium, Austria, Greece, and Malta, households with higher consumption levels could achieve mid-level mitigation results.

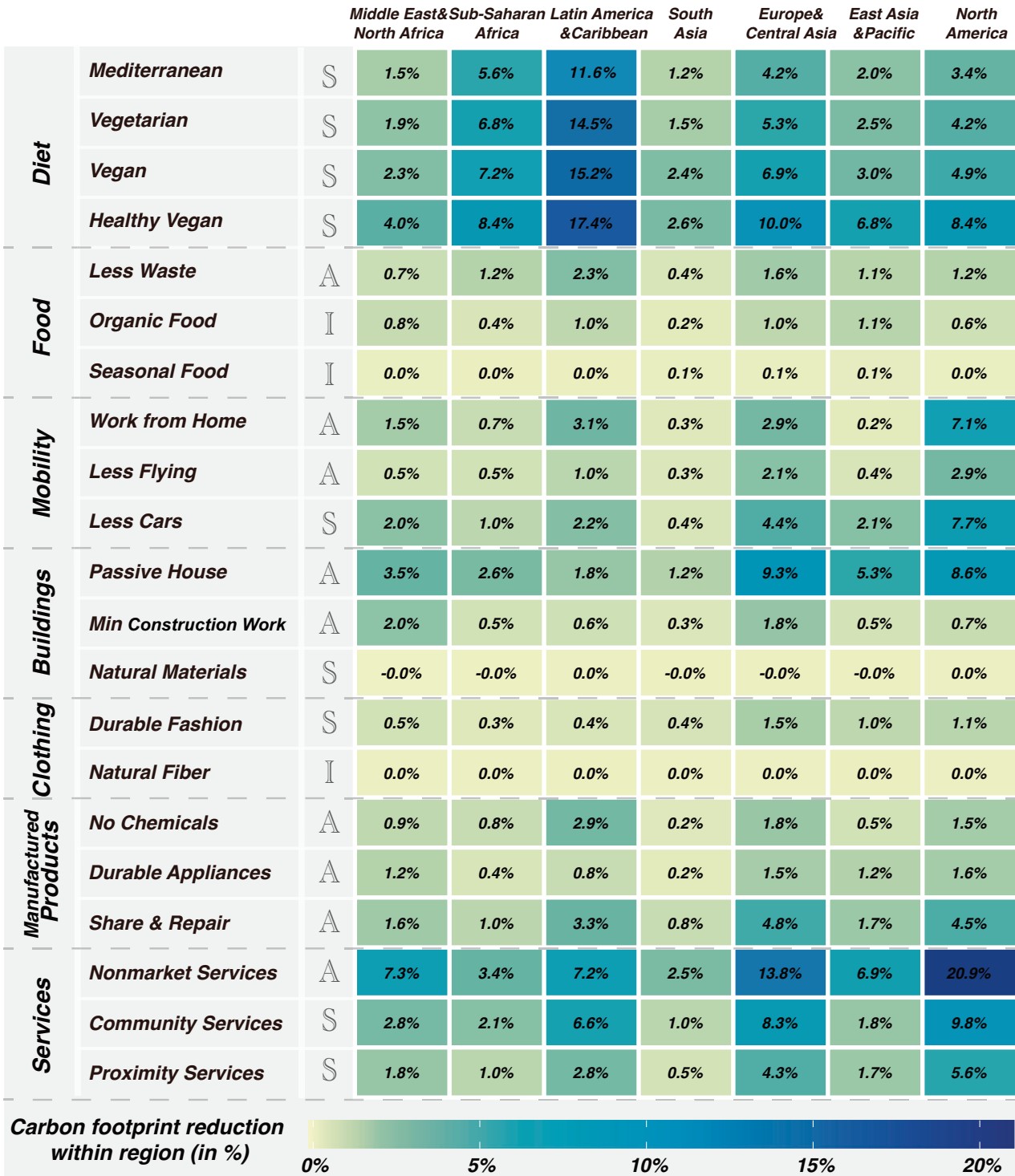

| | | | Middle East&<br>North Africa | Sub-Saharan<br>Africa | Latin America<br>&Caribbean | South<br>Asia | Europe&<br>Central Asia | East Asia<br>&Pacific | North<br>America |
|---|---|---|---|---|---|---|---|---|---|
| Diet | Mediterranean | S | 1.5% | 5.6% | 11.6% | 1.2% | 4.2% | 2.0% | 3.4% |
| | Vegetarian | S | 1.9% | 6.8% | 14.5% | 1.5% | 5.3% | 2.5% | 4.2% |
| | Vegan | S | 2.3% | 7.2% | 15.2% | 2.4% | 6.9% | 3.0% | 4.9% |
| | Healthy Vegan | S | 4.0% | 8.4% | 17.4% | 2.6% | 10.0% | 6.8% | 8.4% |
| Food | Less Waste | A | 0.7% | 1.2% | 2.3% | 0.4% | 1.6% | 1.1% | 1.2% |
| | Organic Food | I | 0.8% | 0.4% | 1.0% | 0.2% | 1.0% | 1.1% | 0.6% |
| | Seasonal Food | I | 0.0% | 0.0% | 0.0% | 0.1% | 0.1% | 0.1% | 0.0% |
| Mobility | Work from Home | A | 1.5% | 0.7% | 3.1% | 0.3% | 2.9% | 0.2% | 7.1% |
| | Less Flying | A | 0.5% | 0.5% | 1.0% | 0.3% | 2.1% | 0.4% | 2.9% |
| | Less Cars | S | 2.0% | 1.0% | 2.2% | 0.4% | 4.4% | 2.1% | 7.7% |
| Buildings | Passive House | A | 3.5% | 2.6% | 1.8% | 1.2% | 9.3% | 5.3% | 8.6% |
| | Min Construction Work | A | 2.0% | 0.5% | 0.6% | 0.3% | 1.8% | 0.5% | 0.7% |
| | Natural Materials | S | -0.0% | -0.0% | 0.0% | -0.0% | -0.0% | -0.0% | 0.0% |
| Clothing | Durable Fashion | S | 0.5% | 0.3% | 0.4% | 0.4% | 1.5% | 1.0% | 1.1% |
| | Natural Fiber | I | 0.0% | 0.0% | 0.0% | 0.0% | 0.0% | 0.0% | 0.0% |
| Manufactured Products | No Chemicals | A | 0.9% | 0.8% | 2.9% | 0.2% | 1.8% | 0.5% | 1.5% |
| | Durable Appliances | A | 1.2% | 0.4% | 0.8% | 0.2% | 1.5% | 1.2% | 1.6% |
| | Share & Repair | A | 1.6% | 1.0% | 3.3% | 0.8% | 4.8% | 1.7% | 4.5% |
| Services | Nonmarket Services | A | 7.3% | 3.4% | 7.2% | 2.5% | 13.8% | 6.9% | 20.9% |
| | Community Services | S | 2.8% | 2.1% | 6.6% | 1.0% | 8.3% | 1.8% | 9.8% |
| | Proximity Services | S | 1.8% | 1.0% | 2.8% | 0.5% | 4.3% | 1.7% | 5.6% |

**Carbon footprint reduction within region (in %)**

0%  5%  10%  15%  20%

**Fig. 4 | Regional carbon reduction potentials through 21 low-carbon expenditures.** The colours indicate the reduction compared to the baseline (i.e., carbon footprints of 2017 within each region). The darker the colour, the greater the reduction potential. Letters 'A', 'S', and 'I' refer to three low-carbon approaches: avoid, shift, and improve, respectively. Specific numeric results are presented. Overlaps between different expenditures and potential rebound effects are not considered in this figure.

## Offsetting potential reductions through rebound effects

The adoption of low-carbon expenditures can lead to money savings. However, unintended rebound effects, where these saved expenditures are re-spent, can offset the initial mitigation benefits[38]. Our study excludes direct rebound effects, as it is unlikely for consumers to increase their consumption of the same fuel product due to energy efficiency gains within the context of our low-carbon lifestyle scenarios[33,36]. Instead, we focus on indirect rebound effects among final consumers, exploring how saved money is reallocated into consumption of other goods and services and the subsequent rebound in upstream carbon emissions throughout the supply chain.

To estimate these reallocations, a double-semi-log regression model[38,43] is used to calculate marginal expenditure shares across different household groups and countries based on expenditure

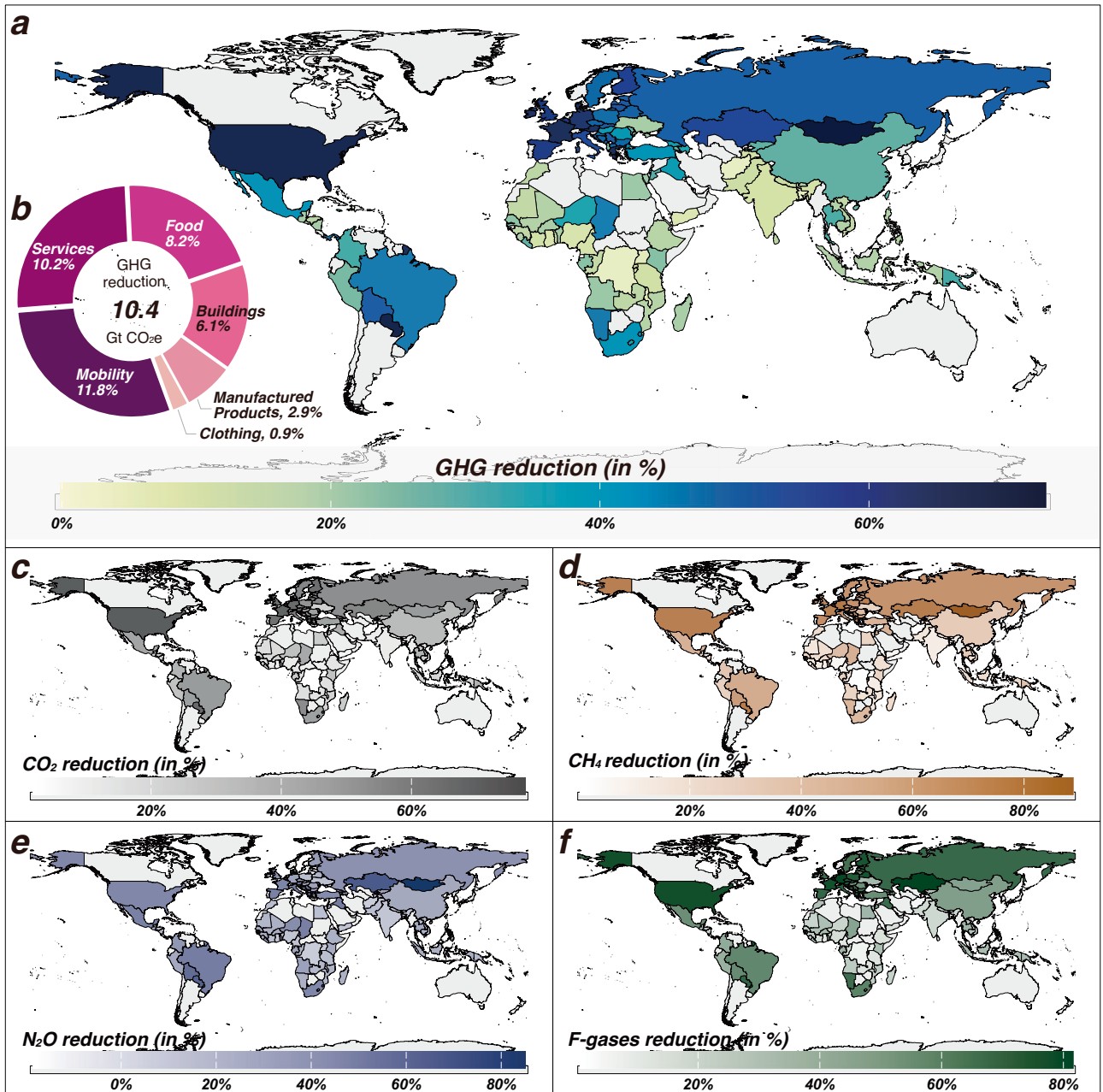

**Fig. 5 | Cumulative carbon savings from a combination of low-carbon expenditures.** The colours of the map in subplots (**a**, **c**–**f**) show relative reduction potentials of GHG, $CO_2$, $CH_4$, $N_2O$, and F-gas compared to their 2017 emissions across 116 countries. The pie chart in subplot (**b**) shows the contributions of changes in product consumption to global GHG reduction. The basemap layer is derived from Runfola, D. et al. geoBoundaries: A global database of political administrative boundaries. PloS one 15, e0231866 (2020), published under the CC BY 4.0 license.

elasticities[36,43,44] (see more details in the *Methods* section). These regression results describe the additional expenditure on a specific commodity for every one-dollar increase in total expenditure, offering a comprehensive insight into how households distribute additional spending across various consumption categories, thereby contributing to the evaluation of potential rebound effects.

Our initial evaluation focuses on rebound effects across 21 distinct low-carbon expenditure measures, where we assumed that all saved money is re-spent on products unaffected by specific low-carbon lifestyle changes, following consumers' marginal expenditure patterns. For example, reduced air travel decreased spending on airfare but left expenditures on other non-aviation goods constant. The freed-up funds from air travel savings were then distributed proportionally (based on their marginal propensity to spend) among these non-aviation travel items and other goods. We found considerable disparities in rebound effects across different lifestyles (see Supplementary Fig. 4). Service-related expenditures tend to have greater backfire effects, as savings from cutting back on leisure activities are often redirected to areas with higher carbon intensity, such as increased spending on food or extended time at home (raising residential energy use).

When exploring the potential magnitude of rebound effects resulting from combined lifestyle changes, we designed various alternative re-spending patterns, taking inspiration from refs. 33,37,38. As shown in Fig. 8, our analysis involves three rebound scenarios (SC1-SC3). SC1 assumes that all saved money is re-spent on products unaffected by low-carbon expenditures. In this case, it is estimated that 4.8 Gt $CO_2e$ GHG reductions are offset, resulting in a rebound effect of

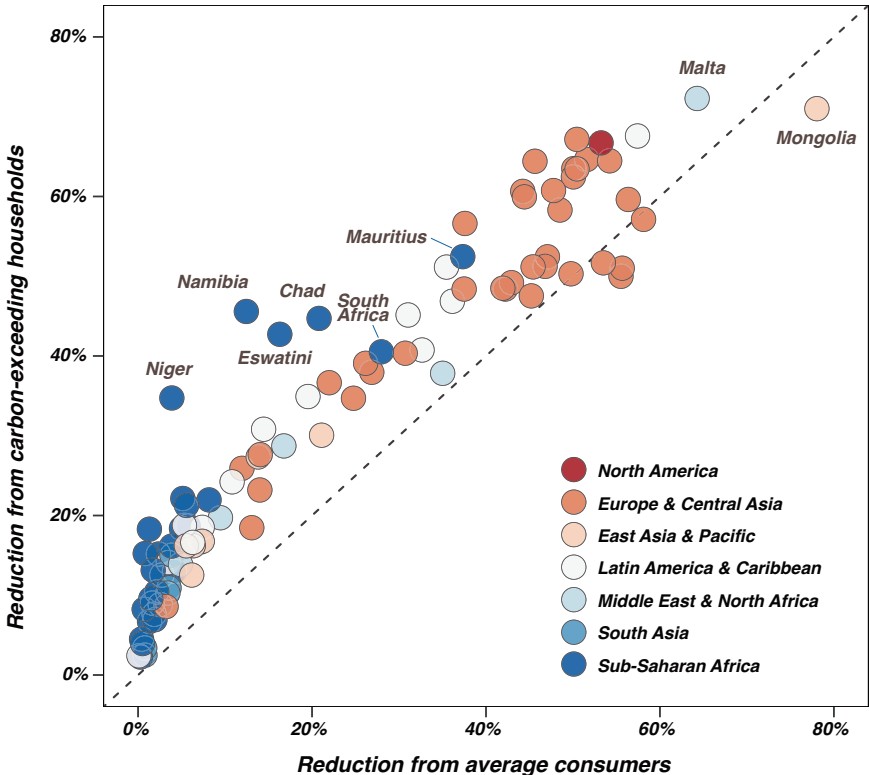

**Fig. 6 | Comparison of national carbon savings through low-carbon expenditures implemented by carbon-exceeding households and randomly selected average consumers.** The y-axis represents national carbon savings achieved by carbon-exceeding households through low-carbon expenditures. The x-axis shows the reductions resulting from applying the same changes to an equivalent proportion of "average consumers" in each country.

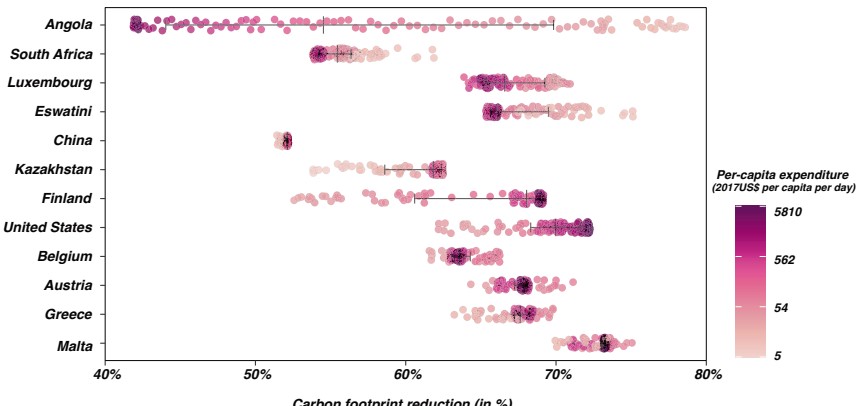

**Fig. 7 | Relative carbon reduction potentials of households from lifestyle changes in selected countries.** The x-axis displays the relative carbon reduction potentials of household groups, indicating the reduction potential of each household group in relation to its 2017 baseline carbon footprint. The median and 25th–75th percentiles (bars) are shown. Sample sizes ($n$) = 116, 114, 95,109, 67, 83, 99, 105, 95, 91, 90, and 112 for Angola, South Africa, Luxembourg, Eswatini, China, Kazakhstan, Finland, United States, Belgium, Austria, Greece, and Malta, respectively. Sample sizes refer to the number of household groups per country; the number of people per household group varies. See Supplementary Fig. 3 for results for all 116 countries studied in this paper.

45.8%. Re-spending on the buildings category leads to a reduction loss of 2.1 Gt $CO_2$e. Drawing inspiration from Druckman et al. [33] and Grabs et al.[38], SC2 and SC3 focus on redirecting all saved money towards the six lowest carbon-intensive products across six consumption categories (SC2) and the least carbon-intensive products among all expenditure items (SC3). While the outcomes of SC2 and SC3 are unlikely in reality, they provide insights into the lower bounds of the rebound effects. Under SC2, we found that 4.5 Gt or 43.8% of the expected global carbon reductions resulting from low-carbon lifestyles are eroded due to such re-spending. Notably, 3.1 Gt $CO_2$e of this loss is induced by re-spending on mobility. Meanwhile, SC3 reflects the best case, indicating that only 0.7 Gt $CO_2$e of carbon savings would be lost, resulting in a rebound effect of 6.5%. SC3 involves the re-spending of saved expenditures primarily on services-related products characterised by lower embodied carbon intensity compared to products in other categories.

We also discovered diverse impacts and potential magnitude of rebound effects experienced by different countries, stemming from

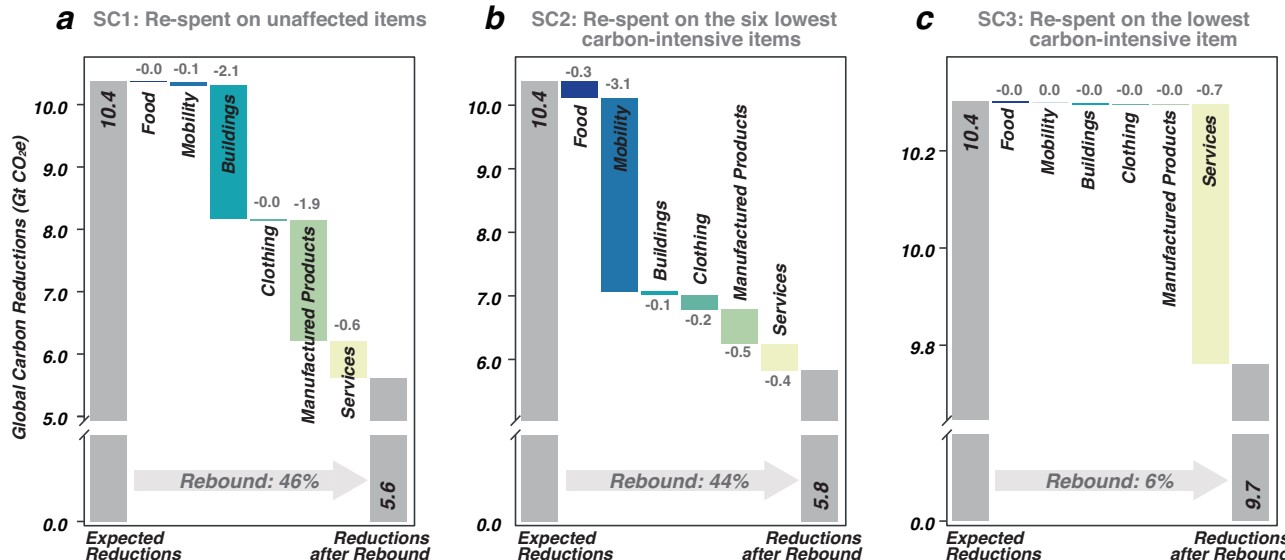

**Fig. 8 | Global carbon reduction offset triggered by rebound scenarios SC1-SC3.** Expected reductions based on lifestyle changes and the reductions after the rebound effect under each scenario are depicted with grey bars in (**a**–**c**). The absolute loss reductions by expenditure categories are highlighted with red numbers. The percentages in the arrow show the rebound effect (defined as the ratio of offsetting carbon reductions resulting from re-spending saved money to expected reductions, see more in the Methods section).

various re-spending patterns (ranging from 12.8% to 228.8% in SC1, 3.8% to 135.0% in SC2, and 0.3% to 17.1% in SC3) (see Supplementary Fig. 5). Due to relatively higher energy costs or the absence of consumption level saturation[32], emerging and developing economies are prone to experiencing more pronounced backfire situations caused by such re-spendings than advanced economies. That underscores the heightened importance of implementing rebound mitigation policies specifically tailored for emerging and developing economies.

## Discussion

This research shows the potential of adopting low-carbon lifestyle changes among high-carbon households, leading to a considerable contribution to GHG emission mitigation at a global scale. Our estimations reveal that implementing a combination of low-carbon lifestyles in 23.7% of the top-emitting population could potentially result in a 40.1% reduction in household consumption-based GHG emissions in the 116 analysed countries (equivalent to 31.7% of the global household carbon footprint in 2017). We observed that nations in North America, Europe & Central Asia, and Latin America & Caribbean exhibit substantial potential for reducing GHG emissions mainly due to their high per-capita carbon footprints and the large number of households involved in our lifestyle-oriented mitigation actions. An interesting finding of this article is the unexpected demand-side mitigation possibilities observed in some countries in Sub-Saharan Africa, such as Mauritius, Namibia, and Chad, which have been overlooked in previous studies.

When exploring emission hotspots among carbon-exceeding populations, we identified key consumption categories with high climate relevance such as buildings and food. Within the buildings category, major contributors to emissions encompass home energy use, construction works, and building material. Our mitigation scenarios highlight the large potential for carbon reductions through low-carbon lifestyle changes, particularly in services, diet, and buildings. Interventions such as reducing food waste, limiting long-distance travel and leisure activities, and shifting towards plant-based diets offer immediate and tangible benefits. These 'low-hanging fruits' represent key opportunities for policymakers to achieve progress with relatively straightforward measures. However, measures targeting the clothing sector, yield only little impact on GHG reductions primarily stemming from the inherently low-carbon-intensive characteristics of the clothing sectors. Thus, while impactful and easily implementable interventions are available, effective emissions reduction requires prioritising the most significant sources and opportunities for change.

The question of how to realise low-carbon lifestyle changes is pivotal, as an individual's choices are intricately linked to income and consumption levels, willingness, resource accessibility, and fiscal and policy frameworks[2]. A combination of regulatory, economic, and information-based instruments, referred to as a "policy package", is generally more effective in achieving these transitions than relying on single policy instruments alone[45]. Governments worldwide have initiated various policies to support lifestyle changes, especially in critical areas of buildings, diet, and mobility. For example, countries such as Spain and Pakistan have promoted shorter working times and encouraged remote work to conserve energy[46]. Investments in transport infrastructure, such as cycle lanes and high-speed rail systems, facilitate a shift towards more sustainable modes of mobility[47]. In response to the energy price crisis triggered by the Russian-Ukrainian conflict, European nations like Germany, the Netherlands, and France, have initiated campaigns aimed at fostering energy-saving actions, including lowering the heating temperature and reducing showers[31,46]. Furthermore, carbon pricing mechanisms, such as taxes and cap-and-trade systems, have proven effective in altering consumer behaviour by incorporating the environmental costs of carbon-intensive goods[48]. Subsidies and tax incentives for renewable energy adoption, alongside information-based interventions such as carbon labelling and customised information feedback[49–51], have facilitated the transition towards greener consumption patterns. The effectiveness and feasibility of these policies are likely to vary across different countries and income groups. In high-income countries, where infrastructure and resources are already in place, more aggressive measures may be viable[52]. Similarly, low-income countries should avoid investing in carbon-intensive infrastructures that would lock them into high carbon-intensive expenditures[53]. For high-income groups, policies could focus on curbing the consumption of luxury goods with high carbon footprints, potentially through progressive taxation or incentives for adopting low-carbon alternatives[39,48]. Conversely, in lower-income regions, the

priority might be on improving access to affordable and sustainable options, ensuring that low-carbon lifestyle changes do not exacerbate existing inequalities[7,54].

Implementing low-carbon lifestyle changes comes with its own set of challenges, particularly the risk of unintended consequences such as rebound effects, where the cost savings from adopting low-carbon lifestyles could lead to increased consumption elsewhere, partially offsetting carbon reductions. Depending on the re-spending scenarios employed in this study, the estimated indirect rebound effects range from 6.5% to 45.8% of the expected global carbon reductions, potentially offsetting carbon savings between 0.7 Gt $CO_2$e and 4.8 Gt $CO_2$e. Addressing the rebound effects linked to lifestyle changes involves suggested pathways such as improving energy efficiency across various sectors, reducing consumption, and transitioning to greener consumption patterns[32,55]. While energy efficiency improvements are effective, they are constrained by current technologies and require substantial investments. Downsizing carbon-intensive consumption appears immediately accessible, particularly to wealthier populations due to their financial security and capacity to forgo non-essential goods and services. However, in the long term, aligning policies aimed at reducing demand with the existing GDP-based economic growth paradigm poses a challenge[56,57]. This study emphasises promoting frugal behaviours among higher-income groups within the context of consumption sufficiency. Additionally, encouraging consumers to re-spend saved expenses on relatively low-carbon-footprint products could help alleviate rebound effects. Examples include purchasing environmentally friendlier electronic products and smart home devices and opting for sustainable tourism practices, such as low-carbon vacations or eco-tourism[58,59].

In summary, low-carbon lifestyles can play a pivotal role in short and medium-term climate mitigation efforts by reducing energy demand and overall consumption. These actions offer a swift and effective means of curbing climate change and carry fewer environmental risks compared to the implementation of technology-based measures[1]. Recent events like the COVID-19 pandemic and the global energy crisis triggered by the Russia-Ukraine conflict have demonstrated that rapid, widespread, and profound changes in lifestyles are possible with government and civil society coordination[46]. However, we have to recognise that such lifestyle changes primarily through reducing household demand could lead to rebound effects due to re-spendings elsewhere. Policymakers should pay attention to mitigating such unintended consequences when designing carbon-saving policies. Additionally, achieving substantial, lasting emissions reductions requires both demand-side measures and supply-side technology solutions. They provide complementary solutions rather than one being superior to the other. Implementation of supply-side technologies faces technical challenges and may require significant timing and investment. In this case, demand-side solutions provide breathing space for the deployment of long-term technology-based reduction measures. Finally, while this study focuses on consumption-based mitigation, consumer choices are largely influenced by availability and accessibility. To achieve meaningful and widespread adoption of low-carbon lifestyles, policies must also address the production side, ensuring that sustainable options are both widely available and affordable.

## Methods
### Overview
In this paper, we used an environmentally extended multi-regional input-output (EEMRIO) approach to estimate household carbon footprints including both direct GHG emissions from home and private transport fuel use and indirect GHG emissions due to fossil fuel uses throughout global supply chains. Subsequently, we modelled various low-carbon expenditures (reflected in changes in household direct fossil fuel use, household final demand and intermediate industrial inputs) with EEMRIO, thereby quantifying the potential impacts of low-carbon lifestyles on carbon reductions.

In alignment with climate justice principles, our lifestyle changes focused on carbon-exceeding households worldwide—those exceeding the global average carbon targets required to stay below 2 degrees—as these populations have contributed most to climate change and possess the greatest capacity for emissions reductions[7–9]. We then assessed the aggregated mitigation potentials by combining multiple low-carbon expenditure measures. We also designed three alternative re-spending scenarios to investigate the potential magnitude of rebound effects resulting from combined lifestyle changes. Data sources and processing are provided. Assumptions and limitations for all calculations are also given.

### Data sources and processing
To model the impacts of changes in expenditures/consumption patterns on carbon emissions, we applied the following main data sources and preparation steps. All other socioeconomic data (for example, population and GDP in PPP (constant 2017 international dollars) used in this study were obtained from the World Bank[60]. The classification of countries by income follows the World Bank[25]. The categorisation of advanced economies and emerging and developing economies is based on the International Monetary Fund[11].

We first linked global MRIO data with detailed household expenditure data. The original global MRIO table is sourced from the Global Trade Analysis Project (GTAP) 11 Database (pre-release version), offering detailed information on interregional and intersectoral transactions of the world economy in 2017. It encompasses 160 economies (141 countries and 19 aggregated regions) with 65 economic sectors each[61]. GTAP has only one vector for household final consumption for each country. To better capture different expenditure patterns within each country, we used a modified household expenditure database derived from the World Bank's Global Consumption Database (WBGCD)[18,62,63]. The modified WBGCD covers detailed consumption information across 33 categories of consumption items and 201 expenditure levels (i.e., expenditure groups) spanning 116 countries. This accounts for 87.3% of the global population and 79.5% of the global GDP, with substantial representation from emerging and developing economies. These datasets are the most detailed available to date. We then developed a bridging matrix to link these 33 consumption categories to the 65 sectors in the GTAP MRIO table, following our previous studies[8,9,31]. Using this matrix, we calculated the consumption shares for each sector by expenditure groups for the 65 sectors in the GTAP. We focused on using the expenditure shares from the WBGCD instead of absolute expenditure values provided by the global consumption database. This approach yielded household final demand data that aligns with the GTAP sectoral classification and maintains consistency across various consumption segments while preserving the detailed differentiation among expenditure groups offered by the WBGCD. Our finalised household final consumption vectors cover 201 expenditure groups for each of the 116 WBGCD countries, across 65 economic sectors within the 160 GTAP countries and regions. The Supplementary Data 3 outlines the 116 countries analyzed in this study.

Production-based GHG emissions (PBE) data (excluding emissions from land-use change) for 2017 are obtained from the GTAP 11 Database (official Release)[61]. PBE cover carbon dioxide ($CO_2$), methane ($CH_4$), nitrous oxide ($N_2O$), and fluorinated gases (F-gases) from fossil fuel combustion, non-energy use, industrial processes, and agriculture activities in 160 countries and regions, divided into 65 sectors. Household direct $CO_2$, $CH_4$ and $N_2O$ emissions were further categorised into residential (heating, cooling, cooking) and private transportation components based on 2017 energy consumption data from the International Energy Agency (IEA) World Energy Balances[64].

The low-carbon expenditure scenarios dataset used in this study originate from Vita et al.'s study[16], which is an outcome of the GLA-MURS project funded by the European Commission's 7th Research Framework Programme. This project adopts an approach centred around envisioning desirable future scenarios, identifying pertinent policies, and conducting backcasting scenario workshops across seven EU regions. The final dataset, integrated with EXIOBASE MRIO analysis, encompasses 36 distinct sufficiency and green consumption behaviours, their corresponding change rates in household final demand and/or industrial inputs, and average uptake rates across these behaviours (without differentiation between countries). Within our study's framework, we intentionally selected 21 low-carbon expenditures from this dataset that align with our modelling objectives. These selections cover seven consumption categories: food, diet, mobility, buildings, clothing, manufactured products, and services. This exclusion process involved omitting less relevant behaviours, such as only consuming local food as it is unrealistic on a global scale. Our analysis uses Vita et al.'s study[16] on the potential for demand changes in households and industries for specified products and the penetration rates of these changes across households and industries. Furthermore, while Vita et al.'s study[16] is intended for integration with the EXIOBASE modelling, an outstanding MRIO dataset primarily centred on European countries, our research objectives suggest a broader scope. We assert that super-emitting households in emerging and developing economies also possess substantial potential for carbon mitigation[7,8,48]. To fully explore this potential, we integrated the GTAP MRIO dataset and aligned the 21 low-carbon expenditures with GTAP's consistent sectoral classification. This strategic alignment enables us to effectively capture and quantify the considerable carbon mitigation potential within these specific households. We listed the 21 low-carbon expenditures and associated modelling parameters used in this study in Supplementary Data 2.

### Baseline household carbon footprints

We used an EEMRIO framework to calculate the household carbon footprint (**CF**) as a baseline (year 2017) and then compared it with the resulting footprints from the low-carbon expenditures. EEMRIO analysis is the most widely used approach for carbon footprinting especially for global studies[7,8].

As shown in Eq. (1), **CF** is a row vector of the total carbon emissions or carbon footprints, encompassing both household direct carbon emissions (**CF**$^d$) from fossil fuel use in homes and private transport, as well as indirect emissions ($\varepsilon(I-A)^{-1}\widehat{\mathbf{Y}}$) associated with household consumption of various goods and services, generated upstream along global supply chains.

$$\mathbf{CF} = \boldsymbol{\varepsilon}(I-A)^{-1}\widehat{\mathbf{Y}} + \mathbf{CF}^d \qquad (1)$$

where $\boldsymbol{\varepsilon}$ is a row vector of carbon coefficient (i.e., carbon emissions per unit of economic output for all sectors in all countries). $L = (I-A)^{-1}$ is the Leontief inverse matrix, which refers to the total inputs of sectors in countries that are satisfied by the outputs of another. $A$ refers to the technological coefficient matrix, with its elements representing the intersectoral economic linkages between the regions. $I$ is an identity matrix with the same size as $A$. $\mathbf{Y}$ is the column vector of final demand. As we focus on the changes in household consumption patterns, thus, the carbon emissions by government consumption and investments are not included in the study. Therefore, **Y** covers only household consumption.

To link lifestyle interventions into environmental analyses, widely used methods include life cycle assessment[65], input-output analysis[16,66], computable general equilibrium models[67] and integrated assessment models[45]. The IO framework, especially global multiregional IO, stands out as a tool for evaluating the mitigation potentials throughout the global supply chains[7,8]. Our paper aims to assess

demand-side reduction measures. In this context, EEMRIO is a suitable approach for this study to model the consumption pattern changes among detailed sectoral expenditure information and consider the entire global value chains with respective production patterns and energy mix for each country, globally.

In the data processing phase, we translated parameters in our low-carbon expenditure scenarios dataset into variables of the EEMRIO model. For example, 'Less waste' is translated as a 12% reduction in household spending on all food products, as indicated in our low-carbon expenditure scenarios dataset[16]. This reduction is interpreted as decreasing household final consumption for food-related activities by 12% and applied in the EEMRIO modelling subsequently. In line with the 'avoid-shift-improve' framework[1,12], existing studies[16,66] suggest that changes in individual lifestyles impact either their consumption patterns or production recipes in at least one sector. In this study, we consider the effects on changes in household direct carbon emissions (**CF**$^d$), household final consumption (**Y**), and production recipes ($A$), contributing to a comprehensive analysis of the environmental implications of low-carbon lifestyle interventions.

### Changes in household direct carbon emissions

Household direct emissions were modified by directly reducing household direct energy use (**CF**$^{d,\,red}$). In certain cases, direct emissions may increase (**CF**$^{d,\,en}$) due to the increased energy demand in other aspects. For instance, 'work from home' can decrease household direct energy use for private transport but may increase home fuel use. We can obtain new household direct emissions (**CF**$^{d'}$) as Eq. (2).

$$\mathbf{CF}^{d'} = \mathbf{CF}^d - \mathbf{CF}^{d,\,red} + \mathbf{CF}^{d,\,en} \qquad (2)$$

We made changes associated with residential fuel use and private mobility separately. Equation (3) shows the process for computing the reduced direct emissions (**CF**$^{d,\,red}$).

$$\mathbf{CF}^{d,\,red} = \mathbf{CF}^d \odot (q^d \times t) \qquad (3)$$

where the symbol $\odot$ refers to element-wise multiplication. $q^d$ is the feasible technical reduction potential, with values ranging from 0 to 1, obtained from a dataset derived from household surveys, expert insights, and behaviour analyses[16]. The term $t$ is the assumed penetration rate, which is tailored to each country based on the proportion of population whose carbon footprints exceeding the carbon thresholds. These thresholds are determined based annual target of household carbon footprints. These targets align with 2-degree-consistent pathways outlined by integrated assessment model scenarios[40]. In 2017, approximately 23.7% of the global population, equivalent to 1.6 billion individuals, exceeded the upper target set for 2020, which stands at 4.6 t $CO_2$e per capita per year.

The two parameters $q^d \times t$ reflect the disparity between the intended intervention and the actual adoption rate. For instance, opting to work from home could result in a 50% decrease in private mobility fuel use ($q^d = 0.50$). We individually modelled such changes for each household group in each country. For carbon-exceeding household groups, $t = 1$, which means such households fully adopt the lifestyle change, whereas for carbon-compliant household groups, $t = 0$. The actual reduction is therefore a weighted sum of these individual adjustments.

For specific lifestyle changes that could trigger increased direct energy use, we can compute the increased direct emissions using Eq. (4).

$$\mathbf{CF}^{d,\,en} = \mathbf{CF}^d \odot (q^{en} \times t) \qquad (4)$$

where $q^{en}$ refers to the potential increase attributable to upswings in residential fossil fuel use. For example, revisiting the 'work from home'

action, increased time spent at home correlates with a 20% rise in home fuel consumption ($q^{en} = 0.20$).

## Changes in household final consumption

Within our low-carbon expenditure measures, household final consumption was modified by both directly reducing final consumption ($\mathbf{Y}^{red}$) and increasing it due to the substitution effect ($\mathbf{Y}^{sub}$). The new household final consumption vectors can be calculated as Eq. (5):

$$\mathbf{Y}' = \mathbf{Y} - \mathbf{Y}^{red} + \mathbf{Y}^{sub} \tag{5}$$

Equation (6) shows the process for computing the reduced final demand ($\mathbf{Y}^{red}$).

$$\mathbf{Y}^{red} = \mathbf{Y} \odot (\mathbf{q}^y \times t) \tag{6}$$

where the column vector $\mathbf{q}^y$ represents the household technical potential for final consumption of products from different countries, with values of elements ranging from 0 to 1. $\mathbf{q}^y \times t$ also reflects the disparity between the intended intervention and the actual adoption rate.

In terms of lifestyle change measures involving shifting effects, where substitution effects occur, the sum of all elements in the row vector $\mathbf{Y}^{red}$ is substituted by other specific products. For example, when shifting to a vegetarian diet, plant-based food, dairy, and eggs are substitutes. To model that substitution, we employed Engel curves, a method widely used in the literature[38,44]. While both marginal expenditure shares (or marginal propensity to spend) and income elasticities are commonly used, we chose to use marginal expenditure shares due to their methodological simplicity. This approach allows us to describe the additional expenditure on a specific commodity for every dollar increase in total expenditure, providing a clear and straightforward means of estimating the re-spending patterns associated with income effects.

We first performed a double-semi-log regression of subcategory expenditures on total expenditures using our bridged household expenditure data[38,44]. The double-semi-log has been demonstrated to be preferable as it aligns with consumer behaviour and provides the best fit for household expenditure data due to its flexible form ref. [44]. We classified global household per-capita expenditure data (covering 116 countries, 201 expenditure groups and 65 expenditure items per country) into 13 country groups based on geographic region and country income level (refer to Supplementary Data 3 for details). Each country group's 201 expenditure groups are further divided into 4 sets based on their expenditure levels. This results in 52 sets $j$ ($j = 1, \ldots, 52$), corresponding to 13 country groups with 4 sets per group. Each set contains 65 expenditure products $i$ ($i = 1, \ldots, 65$). The expenditure of product $i$ within set $j$ can be calculated by $y_{ij}$, as Eq. (7):

$$y_{ij} = \alpha_i + \beta_i {}^* y_{tot,j} + \gamma_i {}^* ln\left(y_{tot,j}\right) + \delta_{ij} \tag{7}$$

where $\alpha_i$ refers to the intercept term. $y_{tot,j}$ denotes the total expenditure of all expenditure items ($\sum_i y_{ij}$). $\beta_i$ and $\gamma_i$ define the relationship of $i$-specific expenditure to total expenditure. $\delta_{ij}$ represents the error term. We then calculated marginal expenditure share ($\tau_i$), as Eq. (8).

$$\tau_i = \beta_i + \frac{\gamma_i}{y_{tot,j}} \tag{8}$$

To express the increased final demand for each household consumption vector, the element $y_k^{sub}$ in $\mathbf{Y}^{sub}$ of product $k$ can be given as

Eq. (9):

$$y_k^{sub} = \begin{cases} p \cdot \mathbf{1} \cdot \mathbf{Y}^{red} \cdot \frac{\tau_k}{\sum_k \tau_k}, & \text{if } k \text{ is a substitutes} \\ 0, & \text{otherwise} \end{cases} \tag{9}$$

where $\mathbf{1}$ is a column vector consisting of all 1 s. The matrix multiplication $\mathbf{1} \cdot \mathbf{Y}^{red}$ allows us to obtain the summation of all elements in $\mathbf{Y}^{red}$. $p$ is the relative price difference between the reduced products and substitutes. $\frac{\tau_k}{\sum_k \tau_k}$ denotes the proportion of final consumption accounted for by product $k$ relative to the total final consumption of all substitutes.

## Changes in production recipes

Certain low-carbon expenditure measures entail alterations to the $A$ matrix (the size is $m \times m$), achievable by reducing intermediate inputs and/or transferring requirements to other intermediate inputs. For instance, opting for natural fibres like wool, fur, cotton, and leather can be represented mathematically within our model by decreasing the intermediate inputs of plastic and rubber products to sectors of textile production and wearing apparel, and substituting them with natural fibres. As Eq. (10), the new matrix ($A'$) can be calculated by:

$$A' = A - A^{red} + A^{sub} \tag{10}$$

We partition the coefficients matrix as $A = [\mathbf{a}_1, \cdots, \mathbf{a}_m]$, and the changes are applied to each column vector $\mathbf{a}_n$. The reduced intermediate inputs can be calculated by Eq. (11):

$$\mathbf{a}_n^{red} = \mathbf{a}_n \odot \left(\boldsymbol{q}^A \times t^A\right) \tag{11}$$

where the column vector $\boldsymbol{q}^A$ refers to the potential reduction rate achievable for each product sector and $t^A$ is the associated assumed industrial penetration rate for each county. We determined $t^A$ by assessing the expenditure of individuals adopting low-carbon lifestyles as a fraction of the total national expenditure of each country.

The substitution of inputs is then applied to the products $k$ as Eq. (12):

$$a_{k,n}^{sub} = \begin{cases} p \cdot \mathbf{1} \cdot \mathbf{a}_n^{red} \cdot \frac{a_{k,n}}{\sum_k a_{k,n}}, & \text{if } k \text{ is a substitutes} \\ 0, & \text{otherwise} \end{cases} \tag{12}$$

where $a_{k,n}^{sub}$ refers to the elements of vector $\mathbf{a}_n^{sub}$.

## Mitigation potential assessment

After that, we can calculate new carbon footprints $\mathbf{CF}' = \boldsymbol{\varepsilon}(I - A')^{-1}\hat{\mathbf{Y}}' + \mathbf{CF}^{d'}$. Thus, expected carbon reductions can be given by Eq. (13):

$$\Delta\mathbf{CF}^e = \mathbf{CF} - \mathbf{CF}' \tag{13}$$

We first conducted separate assessments for the mitigation potential of 21 low-carbon expenditures. Subsequently, we assessed the cumulative mitigation potentials resulting from combining low-carbon expenditures through the simultaneous implementation of all changes in both $A$ and $\mathbf{Y}$. Specifically, we assumed widespread adoption of all low-carbon expenditures by global carbon-exceeding households. To prevent double counting, we carefully accounted for overlapping impacts, particularly when multiple actions targeted similar household activities. For example, if an individual fully adopts a vegan diet, thereby eliminating meat consumption, and reducing food waste—which similarly involves cutting down on meat consumption—the combined impact of these changes cannot simply be added

together. Additionally, we excluded actions unlikely to yield emission reductions, such as the use of natural building materials in certain countries.

## Modelling indirect rebound effects

In our model, adopting low-carbon lifestyles can lead to money savings ($\Delta y = \mathbf{1} \cdot (\mathbf{Y} - \mathbf{Y'})$), which consumers would re-spend. This phenomenon, known as the rebound effect, involves reallocating saved money into final consumption, potentially leading to increased upstream carbon emissions throughout the supply chain[32,33,44]. Our study focuses on indirect rebound effects, where saved money is redirected into other goods and services[36,38]. Direct rebound effects, involving increased consumption of the same fuel products due to energy efficiency gains, are considered unlikely in the context of low-carbon lifestyle changes[33,36]. Thus, the increased household consumption due to re-spending behaviour from money savings ($\Delta y$) can be calculated by Eq. (14):

$$y_i^{respend} = \begin{cases} \Delta y \cdot \frac{\tau_i}{\sum_i \tau_i}, & \text{if } i \text{ is a respendable product} \\ 0, & \text{otherwise} \end{cases} \quad (14)$$

Afterward, we calculated offsetting reductions through re-spendings ($\Delta \mathbf{CF'}$) using the increased household final demand vector ($\mathbf{Y}^{respend}$), as Eq. (15).

$$\Delta \mathbf{CF'} = \boldsymbol{\varepsilon}(I - A')^{-1} \widehat{\mathbf{Y}^{respend}} \quad (15)$$

Actual carbon savings after re-spendings can be given by $\Delta \mathbf{CF}^a = \Delta \mathbf{CF}^e - \Delta \mathbf{CF'}$. Therefore, the rebound effect is expressed as a percentage of offsetting carbon emissions to the expected carbon savings, as Eq. (16):

$$R = \frac{\Delta \mathbf{CF}^e - \Delta \mathbf{CF}^a}{\Delta \mathbf{CF}^e} = \frac{\Delta \mathbf{CF'}}{\Delta \mathbf{CF}^e} \quad (16)$$

## Modelling alternative re-spending scenarios

To assess the potential size of rebound effects resulting from lifestyle changes, we developed three alternative re-spending/rebound scenarios (SC1-SC3), drawing inspiration from studies[33,37,38].

SC1 assumed that all saved money is re-spent on products whose expenditures remain unaffected by low-carbon lifestyle changes[66]. For example, cutting down on air travel could reduce spending on aviation travel items like plane tickets, while expenses on other non-aviation travel items and other goods remain unchanged. The saved money from reduced airfare would then be redirected towards spending on these non-aviation travel items and other goods.

This scenario is grounded in the idea that individuals might channel their remaining savings into goods and services that are not influenced by interventions[32,36]. In line with the approaches of Druckman et al.[33] and Grabs et al.[38], SC2 is designed to redirect all saved money towards the six least carbon-intensive items across six consumption categories. SC3 focuses on channelling the saved money specifically into the least carbon-intensive products among all expenditure items. Here, we used embodied carbon intensity in the baseline (i.e., $\boldsymbol{\varepsilon}(I - A)^{-1}$) as our focus covers all upstream emissions. It's important to note that while the outcomes of SC2 and SC3 may not precisely reflect real-world scenarios, they serve as valuable tools for understanding the potential lower bounds of rebound effects.

## Limitations

When modelling lifestyle changes using EEMRIO, aligned with existing research[33,66], our study sought to estimate potential ranges of carbon reductions rather than provide precise predictions. Therefore, we assumed the widespread adoption of low-carbon lifestyles by the top 23.7% of the global high-emitting population. We refrained from prescribing specific measures for achieving these lifestyles. Instead, we explored potential carbon savings resulting from changes in consumption patterns driven by the adoption of low-carbon practices. Our model does not incorporate changes in carbon intensity due to exogenous technological advancements or new technology applications; rather, our focus centres on lifestyle-induced shifts in household consumption patterns. Additionally, we acknowledge that not explicitly incorporating price changes and policy nudges limits the model's ability to fully capture consumer behaviours[68–70]. However, modelling such behaviours lies beyond the scope of our current study, which focuses on estimating potential carbon savings under hypothetical redistribution scenarios.

In terms of data and processing, our study faced limitations due to the availability of relevant datasets. The Low-carbon expenditures dataset primarily originated from Vita et al.'s study[16], with additional revisions based on refs. 66,71. The raw dataset is focused on the entire population of European countries without country-specific breakdown. We applied it to 23.7% of the global top-emitting population, assuming similar penetration abilities for lifestyle changes as observed in the European populations. While this poses a limitation, the raw dataset remains the most detailed and suitable for lifestyle-oriented MRIO studies currently available. Another limitation is that the GTAP 11 Database and household expenditure survey dataset from the WBGCD lack coverage for all countries and recent-year data are unavailable, as described in our previous studies[8,31]. Nevertheless, they represent the most detailed datasets currently accessible. For the bridged household consumption data, households were grouped based on expenditure levels, without explicitly considering additional indicators that influence household consumption, such as age, household size, and temperature. Another limitation of the model is its reliance on household expenditure surveys combined with EEMRIO Analysis, which may overestimate the carbon footprint of wealthy individuals and consequently overestimate potential reductions[42,72]. This is due to the simplistic assumption that spending in a category directly correlates with carbon emissions, while in reality, there is considerable heterogeneity among products and suppliers within each category[41,42,72]. To address this, we conducted a verification analysis, detailed in the Supplementary Methods.

In designing re-spending scenarios, as we mentioned before, while our scenarios may not precisely reflect real-world situations, they serve as valuable tools for understanding the potential lower bounds of indirect rebound effects. Our study excludes direct rebound effects, as consumers increasing their consumption of the same fuel product due to energy efficiency gains within the context of our low-carbon expenditure scenarios is deemed unlikely[33,36]. We did not address macroeconomic effects related to market price, composition of the economy, and economic growth, which might have influenced outcomes given the scale of the assessed policies[32]. However, this contributes to a clearer understanding of the mechanisms of the effectiveness of individual interventions[66]. We refrained from treating money savings as an investment, even though people in many countries have high saving rates, as our current research scope is within household consumption.

## Reporting summary

Further information on research design is available in the Nature Portfolio Reporting Summary linked to this article.

# Data availability

Global MRIO table is sourced from the Global Trade Analysis Project (GTAP) 11 Data Bases (pre-release version), provided by co-author Y.L.

GHG emissions data were obtained from the GTAP 11 Data Bases (official Release)[61]. The global expenditure data can be collected from the World Bank[62]. All other socioeconomic data (for example, population and GDP) used in this study were obtained from the World Bank[60]. The classification of countries by income follows the World Bank[25]. The categorisation of advanced economies and emerging and developing economies is based on the International Monetary Fund[11]. The main results data generated in this study are provided in the main text and Supplementary Data 1-4. More detailed results are available from the corresponding author on request.

## Code availability

The MATLAB scripts of this work are publicly available on Zenodo (https://doi.org/10.5281/zenodo.13618883).

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

## Acknowledgements

This work is supported by National Natural Science Foundation of China (Project no. 72361137002 (K.H.), 72125010 (Y.L.), 72243011(Y.L.)), Horizon Europe Project EU-CHINA-BRIDGE (101137971 (Y.S.)), which are supported by UKRI grant (10132630) at the University of Birmingham, Wellcome Trust (227150_Z_23_Z (Y.S.)), Nederlandse Organisatie voor Wetenschappelijk Onderzoek NOW (482.22.01 (K.H.)), the Fundamental Research Funds for the Central Universities, Peking University (Y.L.) Supported by High-performance Computing Platform of Peking University (Y.L.), and the China Scholarship Council Ph.D. programme (Y.G.).

## Author contributions

Y.G., Y.S., and K.H. designed the research. Y.G. conducted the analysis, developed the model, and performed the research with contributions from Y.H. and Q.N. Y.G. drafted the manuscript with efforts from Y.S. and K.H. Y.L. provided the global multi-regional input-output table. Y.G., Y.S., Y.H., and K.H. participated in the manuscript revision.

## Competing interests

The authors declare no competing interests.
