## [Transparent Peer Review file · Nature Communications]

Unlocking global carbon reduction potential by embracing low-carbon lifestyles

Corresponding Author: Dr Yuli Shan

Version 0:

Reviewer comments:

Reviewer #1

(Remarks to the Author)

This is undoubtedly a very thorough, useful and insightful piece of research. My main criticism (as a social scientist) is the manner in which 'behaviour' and 'lifestyle' are used interchangeably as if they were the same and I would argue they are not. Behaviours are discrete actions that can be observed and measured, and this study emphasises this point. Lifestyle on the other hand is multi-faceted. It consists of constellations of actions and activities driven by individual cognitions or mental processes, and shaped by contextual drivers (such as affordability or income). There is a huge literature in social sciences. In low-carbon research it is more common to see lifestyle as behaviour. So my recommendations are really to acknowledge and be clear on the framing here (see below).

Introduction section

Low carbon lifestyle, definition – please provide a clear definition of lifestyle and how this sits within your framing of 'lifestyle options' and household expenditure.

Introduction Line 32 'Equitable demand side measures'. I understand your work seeks to create a segmentation between high emitters and low and mitigate the former, but don't you think it is equally important (in the pursuance of equity) to also focus on those who are not able to actively take part in a transition of the energy system?

Line 46 – so your segmentation is really about a high emitting lifestyle and a low emitting lifestyle (proxied by behaviour and income)? Would be interesting to speculate about the other drivers of the high emitting groups. Are these always intentional for example?

Line 54 – Lifestyle analysis suggests that certain low-income groups are more likely to engage in certain behaviours, particularly 'Avoid' driven by a need to save money. Are you ignoring the potential from these groups?

Line 60 – 'towards such lifestyles' can you say a bit more about these lifestyles? You are relying on generalisations here, particularly that consumption expenditure is a good proxy for lifestyle. Please defend this stance.

Line 73-83 these are all reductions related to behaviour change, make it clear in the introduction that your conceptualisation and measurement of lifestyle is through 'behaviours and activities'.

Line 91 – 'lifestyle options' be clear. I would argue that lifestyle is not a matter of choice, particularly for low-income groups.

Main Body

Reduction potential from 'choosing' low-carbon lifestyle. Maybe consider rephrasing to 'engaging' in various low-carbon lifestyles. Consider whether lifestyle is really a choice?

Reviewer #2

(Remarks to the Author)

The authors point out that low-carbon lifestyles could reduce CO2 global emissions by around 40%. In this low-carbon

lifestyle, they consider a compendium of lifestyle changes related to food, mobility, buildings, clothing, manufactured products, and services. Within each category, they also consider several options considering avoid, shift, and improve approaches. As compared to previous studies the strength of their analysis is the larger geographical coverage, the inclusion of rebound effects, and the consideration of within-country heterogeneities (particularly related to the income distribution).

In general, the work supports their conclusions and claims. However, I think there are some aspects that should be tackled by the authors.

I think that some of the assumptions made in the study limit the usefulness of the analysis. A cornerstone of their analysis is the inclusion of rebound effects, which adds significant value to the study. Nevertheless, the way these rebound effects are modeled is quite restrictive. In lines 280-281 they write "The saved money from reduced air travel costs would then be redirected towards spending 280 on these non-transportation items". From a consumer perspective, this approach is not very realistic. Goods or services within a given category are normally closer substitutes than across categories. If I want to reduce my carbon footprint I might choose not to go by plane, but I still want to have holidays and will take the train instead. The newly available income from switching from carbon-intensive lifestyle choices is likely to be shared both by goods in the same category and other goods. I believe that excluding goods within the same category is a serious limitation of the study that deserves some discussion in the text.

As far as I understood, another limitation of their study is that they only consider CO₂ emissions. According to Our World in data in 2017 GHG emissions amounted to 52.25 GtCO₂ eq, so, following their 7.9 Gt result, lifestyle changes considering only carbon emissions would reduce global emissions by around 15%. Nevertheless, part of their lifestyle changes considers switching towards less GHG-intensive diets (e.g., vegan or vegetarian). In food production, CO₂ is the least important contributor as CH₄ and NO₂ emissions are normally higher. Hence, this is an important piece missing in their analysis. I think this should be at least discussed in the text and ideally added in their simulations.

For the consumption data, they used the World Bank's Global Consumption Database for 2011. Since the other data they use is for 2017 and the World Bank has already published in 2020 the International Comparison Project dataset of household expenditure for 2017 (International Comparison Program (ICP) - Data (worldbank.org)), I think this data should be considered instead, it also has a larger country coverage. I wonder if the authors have any justification for not using the consumption data that corresponds to the year of their analysis. Could they rerun their analysis with adequate data? I also wonder why they used two different versions of the GTAP 11 Database with different country coverages (141 vs 160 countries). This is puzzling. Finally, they mention in line 395 that they use the World Bank GDP, but which one (nominal, real, in PPP, in Int\$)?

The modeling of household consumption is rather simple. I understand that some simplifications are required when handling larger models. However, since the study concentrates on consumer choices, I think it would be relevant to consider both income effects and price effects, particularly to embed the substitutions and complementarities across different goods, which relates to the comment I made regarding the limitation of the modeling of the rebound effects. If the study focused on production, I would better understand the simplification in the modeling of the demand changes. In the context of this paper, however, further efforts would be welcomed. If they are not undertaken, at least a justification or discussion of these limitations should be added.

In their conclusions, the authors comment on the challenges of promoting low-carbon lifestyle changes in economies attached to the growth paradigm poses a challenge, and then they mention the necessity of promoting frugal behaviors, especially among the higher-income groups. This analysis begs the question of how to achieve these changes. I understand this is beyond the scope of this paper. However, discussing possible policies to promote such lifestyle changes would be a great addition to the study. There is a vast literature of studies simulating the effects of policies promoting several of the lifestyle changes considered. The authors could discuss: (1) What are low-hanging fruits? (2) What is the evidence of different policies for the different goals? (3) Is (Are) there (some) a policy agenda(s) that would make more sense given their results? (4) Does this vary across countries or income groups?

Specific comments:

Line 417 & 91: cite Vita et al.

Reviewer #3

(Remarks to the Author)

The impact of demand-side shifts on global carbon footprints is a critical yet understudied area. This paper contributes significantly by analyzing the potential global impact of lifestyle changes—a relatively novel focus in this field. Although the study does not capture all possible effects, it provides a valuable initial estimate. The results indicate that if the wealthiest 25% of the global population adopted certain lifestyle changes—such as adopting vegan diets, living in passive houses, and reducing car usage—the total carbon footprint could be reduced by a substantial 40%. These changes would undoubtedly have profound economic and societal ripple effects, but as a preliminary exploration, the study's approach is, I believe, justifiable.

One of the main concerns with the manuscript is its reliance on household expenditure surveys combined with Environmentally Extended Input-Output Analysis (EE-IOA). While this methodology aligns well with production-based emissions and is standard for estimating global consumption-based carbon footprints, it may not be entirely appropriate to use "of the shelf" for the purpose of this study. A well-known limitation of EE-IOA models is that they tend to overestimate the carbon footprint of wealthier households due to the simplistic assumption that spending in a category directly correlates with carbon emissions in that category, while in reality there is a large heterogeneity among different products and suppliers. This assumption disproportionately impacts those who spend more on luxury goods and services, potentially skewing results.

This limitation, while often recognized in the literature as a standard caveat in the limitations section, becomes particularly

problematic here because the study's findings hinge on expenditure data from wealthier demographics that might be inflated. Given the study's significant claims about potential emission reductions, this methodological weakness warrants further scrutiny.

Suggested Improvements:

To strengthen their argument, the authors should attempt to cross-verify their estimated impacts with alternative methodologies. For instance, comparing the estimated emissions reduction potential for housing against the total emissions from housing within a country could provide a reality check. Ideally, using physical data to do the same estimation would strengthen the results and enhance credibility. Although research in this area is limited, incorporating findings from studies like Girod de Haan (2010) and André et al. (2024) could provide additional benchmarks to assess the reasonableness of the claims.

Final thoughts:

While I recognize the potential impact of the findings presented in this manuscript, I have significant reservations about the robustness of the results due to the methodological limitations discussed. Previous similar studies, such as those by Vita and Chancel, have acknowledged similar issues as mere limitations. However, in the context of this paper, where such assumptions could substantially influence the primary conclusions, merely noting these as limitations may not suffice. Given the potential overestimation of carbon footprints for wealthier households, which are central to the study's claims, it is crucial to address these concerns more comprehensively. It may be necessary for the authors to explore alternative methodologies or provide more substantial validations for their current approach before considering this study for publication. I have a few other remarks and suggestions, but this is my main concern.

Version 1:

Reviewer comments:

Reviewer #1

(Remarks to the Author)

Thank you for addressing all my review points. I am satisfied that these are adequate and recommend this article for publication.

(Remarks on code availability)

Reviewer #2

(Remarks to the Author)

I would like to thank the authors for taking the time to thoroughly review the paper and seriously consider my suggestions. I think both the inclusion of all GHG and the rewriting of the discussion including possible policy agendas improved substantially the quality of the paper.

I would also like to thank you for clarifying my misunderstandings and for making the text clearer for other readers.

Regarding the reviewed version, I have only a couple of comments left.

First, in line 524 I think you mean that when switching to a vegetarian diet, plant-based foods, and dairy products (and probably also eggs) are substitutes. Using "substituted products" is confusing because this means that they are substituted away. I think you mean that dairy and plant-based products can be used instead of meat in which case the correct word is substitute, as the very pertinent example in the Cambridge dictionary indicates: Tofu can be used as a meat substitute in vegetarian recipes (<https://dictionary.cambridge.org/dictionary/english/substitute>).

Second, I have only one remaining comment regarding the methodology. I don't necessarily think it should lead to a change in the analysis but to an acknowledgment of a limitation and perhaps potential future improvements.

In lines 285-287 you express that the freed-up funds are distributed proportionally (based on their marginal propensity to spend) among the elements of the same category and other goods. The limitation I see in this approach is that the marginal propensity to spend is calculated under current consumption and the y^{sub} is also distributed according to the current consumption. The two elements affecting this substitution are the relative price difference between the two goods and the proportion of final consumption accounted for by substitute k .

You are, however, modeling a change in consumption that in reality goes beyond differences in relative prices and the current consumption share of a good. Say for example that I go vegetarian but before I already ate a large portion of legumes in my diet, since dairy is relatively cheap with respect to meat and I might perceive it more as a closer substitute to meat than legumes, I might substitute almost all the consumption of meat by dairy products and keep my legumes consumption almost constant. This might be a serious concern because if I previously ate mainly poultry and fish, substituting them for dairy products might not lead to the expected emissions reductions. Such consumer preferences are normally modeled by own-price and cross-price elasticities. The insights you can draw from your demand system are constrained by the fact that you do not include such demand elasticities (also called preference parameters sometimes). Therefore I think that this

proportionality based on the marginal propensity to spend is a rather limited way of modelling substitution effects in consumption.

I am aware of what you wrote in line 613, I however do not think it is enough because own-price and cross-price elasticities depend on various socioeconomic aspects, reflecting different price effects on different economies (see Bouyssou, C. G., Jensen, J. D. & Yu, W. Food for thought: A meta-analysis of animal food demand elasticities across world regions. *Food Policy* 122, (2024)). The fact that price effects vary across world regions and income groups is therefore also missing from your analysis and could potentially lead to significantly different results.

In case the authors would like to have a reference of a (also simple) demand system using own- and cross-price elasticities, I think a good example is: Robinson, S. et al. The International Model for Policy Analysis of Agricultural Commodities and Trade (IMPACT): Model Description for Version 3. SSRN Journal (2015) doi:10.2139/ssrn.2741234. That, for example, has been used to model consumption changes stemming from emissions pricing: Springmann, M. et al. Mitigation potential and global health impacts from emissions pricing of food commodities. *Nature Clim Change* 7, 69–74 (2017). Other more complicated demand systems considering such preference parameters are included in the GTAP and other general and partial computable general equilibrium models as is customary in the field of Economics.

For a discussion on the role of preference parameters/demand elasticities in demand systems see:

1. Valin, H. et al. The future of food demand: understanding differences in global economic models. *Agricultural Economics* 45, 51–67 (2014).
2. 1. Ho, M. et al. Modelling Consumption and Constructing Long-Term Baselines in Final Demand. *Journal of Global Economic Analysis* 5, 63–108 (2020).

Summing up, I think the efforts made by the authors are valuable and shed new light on crucial aspects. I would like to recommend it for publication. Nevertheless, I think it is very important to acknowledge the aforementioned limitations on modeling consumer behavior without using preference parameters as they could significantly influence the results and key messages of the paper if they were included.

(Remarks on code availability)

Reviewer #3

(Remarks to the Author)

My main issue with this manuscript remains the potential inflation of results due to the method used. The authors have acknowledged the limitations of the EE-IOA framework and provided a sensitivity analysis, as requested. While I appreciate their efforts, I am not entirely convinced, and I would like to see further clarification in their response.

The EE-IOA framework assumes a linear relationship between spending and carbon footprints, which is reasonable at the aggregate level. However, this assumption becomes problematic when applied to sub-groups that differ significantly from the average, such as high-income households, as examined in this manuscript. These households typically spend more on goods and services, but this does not necessarily reflect their “true” carbon footprint. For example, purchasing a €1000 suit does not generate 100 times the emissions of a €10 T-shirt. The same applies to major investments like housing, where location heavily influences market value, or vehicles, where price often reflects non-emission-related factors. Food and services also show price premiums for elements like branding or expertise that don't correlate with higher emissions. This indicates both price and product heterogeneity that the EE-IOA framework fails to capture. This limitation is widely acknowledged in studies using the EE-IOA framework and, and addressing it in this manuscript could, I believe, contribute to advancing the literature on carbon footprint estimations.

In my first review and as a first point, I suggested that the authors should try to cross-verify their national-level reduction potentials with primary data, such as comparing estimated savings from dietary shifts with FAOSTAT or household energy using EUROSTAT emissions data. While I understand that this would introduce other uncertainties and may be a demanding request, I believe such cross-verification would add valuable validation. The Andre et al. (2024) paper I referenced can be found here: <https://www.sciencedirect.com/science/article/pii/S0921800924000193>.

Requested action: I leave it to the authors to decide how to proceed with this.

The authors propose a sensitivity analysis in the rebuttal, comparing their estimated emission reductions to those obtained if the same changes were made by the corresponding proportion of “average consumers” in each country using national-level MRIO models. Since the average consumer represents the average household emissions, they argue that significant deviations for high-income households would highlight heterogeneity. They find small deviations and conclude that this should not be an issue.

However, I'm not convinced this fully addresses the issue. Even if the relative reduction potentials for both groups appear similar in many cases, this doesn't seem to me to resolve the concern about inflated emissions among high-income households, since it just reflects the relative reduction potential of the already potentially inflated numbers.

Requested action 1: I would appreciate an illustrative example of one or two of these calculations and I think they could also help clarify this section.

Requested action 2: Additionally, I suggest the authors present the total summarized effect of all national-level reduction potential estimated for the average consumers and include it in the article. This estimate would serve as a benchmark

estimate for the effect of the suggested changes, were they to be adopted randomly by the same share of households in each country.

Note: The assumption of 27% reduction from “less waste” seems to have been confused with the measure “Food sufficiency”. The reference estimates the effect of less waste at 12% from what I understood.

Requested action: Look into it.

Comments with no requested actions:

As for modeling alternative re-spending scenarios and specifically the basis for SC3 this empirical article may be interesting: <https://www.sciencedirect.com/science/article/pii/S0959652622053136>

(Remarks on code availability)

Version 2:

Reviewer comments:

Reviewer #2

(Remarks to the Author)

I would like to thank the authors for their replies to my previous comments. I appreciate the change in the modeling of Y_{sub} , which I think is now more realistic.

In the first reply to my comment on the GHG emissions (first round of revisions) the authors wrote:

After recalculating, the global production-based GHG emissions for 160 countries and regions in 2017 amount to 50.56 Gt CO₂e. The global GHG footprints of the 116 countries amount to 32.6 Gt CO₂e in 2017. The total reduction potential from a combination of low-carbon lifestyle measures is 12.8 Gt CO₂e, representing 39.3% of global GHG emissions in 2017.

Given these calculations, I think it is quite misleading to write (in lines 19-20 and 281-282) that: “a combination of low-carbon expenditures among the top 23.7% emitters could cut global carbon footprints by 10.4 Gt CO₂e (i.e., 40.1% of global GHG emissions in 2017)”

If global emissions are 50.56 Gt CO₂e in 2017, then the share of global emissions is 20.6%.

Else, the statement needs to be more precise, e.g., “a combination of low-carbon expenditures among the top 23.7% emitters could cut global carbon footprints by 10.4 Gt CO₂e (i.e., 40.1% of the GHG footprints of the 116 countries we consider in our study in 2017)”.

Also, taking the 32.6 Gt CO₂e in 2017 the authors used before (in the first reply to my comments) and the new estimate of 10.4 Gt CO₂e the percentage becomes 31.9%, not 40.1%. Is there a change in the GHG footprints estimates of the 116 countries you used since the last revision?

I think there is no need to inflate the results by using a different base without being transparent about the base used. A 20.6% of global emissions reduction is a significant number, already, and more accurate and transparent for the reasons I already explained. I hope the authors will understand my concerns about misrepresenting mitigation potentials.

(Remarks on code availability)

Reviewer #3

(Remarks to the Author)

Thank you for your thorough and ambitious response to my previous comments. The additional analyses you have conducted, especially the case studies based on physical data for Switzerland and China, reinforce your findings and rebute my previous concerns. Your willingness to delve deeper is greatly appreciated.

Before I can fully recommend your manuscript for publication, I would like to see a methodological Clarification on Physical Data: It is not entirely clear how you integrated the physical consumption data from your case studies into the emissions calculations. The reference provided does not explain the use of physical data. Providing a more detailed explanation in the supplementary would help readers understand, verify, and potentially replicate your approach here. I realize this is some additional work but ensuring traceability in these calculations is important for building on your work in the future.

I also have a comment on Figure 6: You note that the differences in emission reduction potential between “carbon-exceeding” households and average households are large but the figure might give the reader the impression that it is in fact rather small (looks like the mean difference is 10% larger emission reductions from the carbon-exceeding" group. It might be

worth acknowledging that in wealthier countries, the “carbon-exceeding” group constitutes a substantial portion of the randomly selected cohort. This could explain why the observed differences appear modest and merit explicit mention in the text. Alternatively, think of how this figure could be changed to better reflect your conclusions. This is just a suggestion, do as you please.

If you address the above point, I would be ready to recommend it for publication. You have already made significant efforts to improve the work, and these clarifications will strengthen its overall contribution to the literature.

(Remarks on code availability)

Version 3:

Reviewer comments:

Reviewer #2

(Remarks to the Author)

Thanks a lot for addressing my comments. I think it is now ready for publication.

(Remarks on code availability)

Reviewer #3

(Remarks to the Author)

I'm still unclear whether there is a Chinese HBS survey that collects data in physical units. If so, please provide a specific reference. The current citations aren't sufficient. If the physical data comes only from trade data (e.g., imports/exports in GTAP), it wouldn't address heterogeneity among different household groups.

(Remarks on code availability)

Version 4:

Reviewer comments:

Reviewer #3

(Remarks to the Author)

Ok thank you for this clarification. I have no further comments and recommend publication. Thank you for your work.

(Remarks on code availability)

Response letter to Reviewers' comments

Reviewer 1:

This is undoubtedly a very thorough, useful and insightful piece of research. My main criticism (as a social scientist) is the manner in which 'behaviour' and 'lifestyle' are used interchangeably as if they were the same and I would argue they are not. Behaviours are discrete actions that can be observed and measured, and this study emphasises this point. Lifestyle on the other hand is multi-faceted. It consists of constellations of actions and activities driven by individual cognitions or mental processes and shaped by contextual drivers (such as affordability or income). There is a huge literature in social sciences. In low-carbon research it is more common to see lifestyle as behaviour. So my recommendations are really to acknowledge and be clear on the framing here (see below).

RE: Thank you for your positive comments and suggestions on this paper. We have revised the manuscript according to your feedback. To clarify the usage of 'behaviour' and 'lifestyle,' we have added a paragraph in the introduction section defining our framing. Please see our detailed point-by-point responses below.

Introduction section:

Low carbon lifestyle, definition – please provide a clear definition of lifestyle and how this sits within your framing of 'lifestyle options' and household expenditure.

RE: Thank you for the suggestion. We have added a paragraph in the Introduction to describe low-carbon lifestyles. Please see the revised text below.

To effectively achieve demand-side mitigation, adopting low-carbon lifestyles that minimize GHG emissions is essential^{1,2}. Lifestyle is a multifaceted construct including behaviours, cognitions, and contextual factors^{3,4}. Household consumption expenditure often serves as a reliable proxy for individual lifestyles, reflecting choices across transportation, food, housing, and consumer goods^{5,6}. By examining consumption-related behaviours, such as product selection and usage patterns, we can better understand the drivers of household carbon emissions. For instance, decisions to purchase durable appliances, consume natural fibre clothing or reduce food waste directly impact household carbon footprint. A deeper understanding of these behaviours is crucial for designing effective interventions to promote low-carbon lifestyles.

Introduction Line 32 'Equitable demand side measures'. I understand your work seeks to create a segmentation between high emitters and low and mitigate the former, but don't you think it is equality important (in the pursuance of equity) to also focus on those who are not able to activity take part in a transition of the energy system?

RE: We agree with your point. Climate change mitigation through demand-side measures requires broad and equitable participation from all individuals and communities. While this paper focuses on mitigating the carbon emissions from high emitters, it is equally important to consider those who

may face barriers to actively participating in the transition to a low-carbon economy. We have added an explanation of equitable measures and clarified why our focus is on high emitters. Please see the revised sentences below.

The literature on carbon inequality emphasizes the importance of equitable demand-side measures ⁷. Addressing climate change requires a multifaceted approach that includes cutting emissions from high emitters while supporting those who face barriers to low-carbon transitions, such as energy poverty. Research found that the top 10% of emitters accounted for 48% of the global emissions in 2019, with the top 1% contributing 16.9% to the global total, while the bottom 50% emitted 12% ⁸. Between 1990 and 2019, the bottom 50% contributed only 16% of global emissions growth, whereas the top 1% accounted for 23% of the total ⁸. This disparity highlights that a relatively small, wealthy part of the global population predominantly drives consumption-based emissions ^{9,10}. This situation underscores the urgent need to propose demand-side measures that specifically target carbon-intensive activities among top emitters, as those households have contributed most to climate change and have the greatest capacity for reducing emissions.

Line 46 – so your segmentation is really about a high emitting lifestyle and a low emitting lifestyle (proxied by behaviour and income)? Would be interesting to speculate about the other drivers of the high emitting groups. Are these always intentional for example?

RE: Our segmentation is based on population groups with varying expenditure levels and consumption patterns. Our approach, based on an environmentally extended multi-regional input-output model linked to detailed household expenditure data, allows us to quantify carbon footprints resulting from household consumption activities creating emissions along global supply chains. While we can identify high-emitting groups, determining whether these emissions are intentional or incidental is beyond the scope of this study. Multiple factors, including income and expenditure levels, resource availability, socioeconomic conditions, and individuals' willingness to mitigate climate change, contribute to these disparities ¹¹ as well as differences in production technologies and energy mix along the production chains. This paper scrutinizes hotspots of household carbon footprints in terms of geographical distribution or production and consumption and consumption-related behaviours. In response to this comment, we have expanded the discussion on hotspots of household carbon footprints in the first results section. Please refer to the revised manuscript for details.

Line 54 – Lifestyle analysis suggests that certain low-income groups are more likely to engage in certain behaviours, particularly 'Avoid' driven by a need to save money. Are you ignoring the potential from these groups?

RE: We agree with this point. However, it is important to acknowledge that many low-income populations are still struggling with poverty and hunger. Half of the global population spends less than US\$4.37 per day, and 90% of the globe less than US\$32 ¹². The potential for 'avoid' is much lower than for the global elites with much higher incomes. Our focus on high-emitting groups is because they already have sufficient levels of consumption and are therefore in a better position to transition towards more low-carbon lifestyles.

Line 60 – ‘towards such lifestyles’ can you say a bit more about these lifestyles? You are relying on generalisations here, particularly that consumption expenditure is a good proxy for lifestyle. Please defend this stance.

RE: We have added one paragraph describing low-carbon lifestyles, as our response to your first comment.

Line 73-83 these are all reductions related to behaviour change, make it clear in the introduction that your conceptualisation and measurement of lifestyle is through ‘behaviours and activities’.

RE: Done. We have added one paragraph describing low-carbon lifestyles, as our response to your first comment.

Line 91 – ‘lifestyle options’ be clear. I would argue that lifestyle is not a matter of choice, particularly for low-income groups.

RE: Thanks for your suggestion. We have revised the entire manuscript, replacing “options” with “expenditures” and other relevant terms. Please find the updated manuscript.

Main Body

Reduction potential from ‘choosing’ low-carbon lifestyle. Maybe consider rephrasing to ‘engaging’ in various low-carbon lifestyles. Consider whether lifestyle is really a choice?

RE: Done. We have also double-checked and revised the entire main text to ensure consistency.

Reviewer 2:

The authors point out that low-carbon lifestyles could reduce CO₂ global emissions by around 40%. In this low-carbon lifestyle, they consider a compendium of lifestyle changes related to food, mobility, buildings, clothing, manufactured products, and services. Within each category, they also consider several options considering avoid, shift, and improve approaches. As compared to previous studies the strength of their analysis is the larger geographical coverage, the inclusion of rebound effects, and the consideration of within-country heterogeneities (particularly related to the income distribution). In general, the work supports their conclusions and claims. However, I think there are some aspects that should be tackled by the authors.

RE: Thank you for your positive evaluation of this paper. We have revised the manuscript according to your feedback. Please see our detailed point-by-point responses below.

I think that some of the assumptions made in the study limit the usefulness of the analysis. A cornerstone of their analysis is the inclusion of rebound effects, which adds significant value to the study. Nevertheless, the way these rebound effects are modelled is quite restrictive. In lines 280-281 they write “The saved money from reduced air travel costs would then be redirected towards spending on these non-transportation items”. From a consumer perspective, this approach is not very realistic. Goods or services within a given category are normally closer substitutes than across categories. If I want to reduce my carbon footprint I might choose not to go by plane, but I still want to have holidays and will take the train instead. The newly available income from switching from carbon-intensive lifestyle choices is likely to be shared both by goods in the same category and other goods. I believe that excluding goods within the same category is a serious limitation of the study that deserves some discussion in the text.

RE: Thank you for your insightful comment. We agree with your perspective. There was indeed an error in the wording of our analysis, but our model design and calculations align with your understanding. We assumed that all saved money would be re-spent on products unaffected by specific low-carbon lifestyle measures. The newly available income would be shared both by goods in the same category and other goods based on their marginal propensity to spend. We have revised the example in the main text to better reflect this. Please see the updated sentences below:

Our initial evaluation focuses on rebound effects across 21 distinct low-carbon expenditure measures, where we assumed that all saved money is re-spent on products unaffected by specific low-carbon lifestyle changes, following consumers’ marginal expenditure patterns. For example, reduced air travel decreased spending on airfare but left expenditures on other non-aviation goods constant. The freed-up funds from air travel savings were then distributed proportionally (based on their marginal propensity to spend) among these non-aviation travel items and other goods. We found considerable disparities in rebound effects across different lifestyles (see Supplementary Figure 4). Service-related expenditures tend to have larger rebound effects because savings from cutting back on leisure activities are often redirected to areas with higher carbon intensity, such as spending more on food or increasing time at home (leading to higher residential energy use).

As far as I understood, another limitation of their study is that they only consider CO₂ emissions. According to Our World in data in 2017 GHG emissions amounted to 52.25 GtCO₂ eq, so, following their 7.9 Gt result, lifestyle changes considering only carbon emissions would reduce global emissions by around 15%. Nevertheless, part of their lifestyle changes considers switching towards less GHG-intensive diets (e.g., vegan or vegetarian). In food production, CO₂ is the least important contributor as CH₄ and N₂O emissions are normally higher. Hence, this is an important piece missing in their analysis. I think this should be at least discussed in the text and ideally added in their simulations.

RE: Thanks for the valuable suggestion. We have included all GHG types in the modelling. Please refer to the paragraph on data sources below.

GHG emissions data (excluding emissions from land-use change) for 2017 from both firms and households are obtained from the GTAP 11 Database (official Release)¹³. Firm emissions cover carbon dioxide (CO₂), methane (CH₄), nitrous oxide (N₂O), and fluorinated gases (F-gases) from fossil fuel combustion, non-energy use, industrial processes, and agriculture activities in 160 countries and regions, divided into 65 sectors. Household direct emissions include CO₂, CH₄, and N₂O. Household CO₂ emissions were further categorized into residential (heating, cooling, cooking) and private transportation components based on 2017 energy consumption data from the International Energy Agency (IEA) World Energy Balances¹⁴.

After recalculating, the global production-based GHG emissions for 160 countries and regions in 2017 amount to 50.56 Gt CO₂e. The global GHG footprints of the 116 countries amount to 32.6 Gt CO₂e in 2017. The total reduction potential from a combination of low-carbon lifestyle measures is 12.8 Gt CO₂e, representing 39.3% of global GHG emissions in 2017. Additionally, we found considerable reduction potential from diet-related measures, particularly from CH₄ and N₂O emissions. We have updated the manuscript based on these new results. Please see the revised manuscript for details.

For the consumption data, they used the World Bank's Global Consumption Database for 2011. Since the other data they use is for 2017 and the World Bank has already published in 2020 the International Comparison Project dataset of household expenditure for 2017 (International Comparison Program (ICP) - Data (worldbank.org)), I think this data should be considered instead, it also has a larger country coverage. I wonder if the authors have any justification for not using the consumption data that corresponds to the year of their analysis. Could they rerun their analysis with adequate data?

RE: Thank you for your comment. We have reviewed the household expenditure data in the International Comparison Project dataset, but it only provides country-level data. In contrast, the World Bank's Global Consumption Database, which we used in this paper, offers household-specific expenditure data. In response to your comment, we have revised the data description section to clarify the details regarding our data sources. Please see the paragraph below.

To capture diverse population consumption patterns, we integrated household expenditure data for 2011 from the World Bank's Global Consumption Database (WBGCD)¹², a method consistent with our previous studies^{9,10,15}. The WBGCD covers 116 countries, accounting for 87.4% of the global population and 80.3% of the global GDP, with substantial representation from developing countries. Each country is categorized into up to 201 expenditure groups based on per capita annual consumption levels, ranging from \$0 to \$1 million in 2011 Purchasing Power Parity (PPP) terms. These datasets are the most detailed available to

date. For consistency, we used the detailed expenditure structure (instead of real consumption data) from the WBGCD to disaggregate the total final household demand in the GTAP MRIO table. Our finalized household final consumption vectors cover 201 expenditure groups from 116 WBGCD countries across 65 economic sectors within the 160 GTAP countries and regions. The Supplementary Table 3 outlines the 116 countries analyzed in this study.

I also wonder why they used two different versions of the GTAP 11 Database with different country coverages (141 vs 160 countries). This is puzzling.

RE: Thank you for pointing this out. There seems to be some misunderstanding. The GTAP 11 dataset covers 160 economies, including 141 countries and 19 aggregated regions. We have added further details to the data sources section to clarify this and prevent any confusion.

The original global MRIO table is sourced from the Global Trade Analysis Project (GTAP) 11 Database (pre-release version), offering detailed information on interregional and intersectoral transactions of the world economy in 2017. It encompasses 160 economies (141 countries and 19 aggregated regions) with 65 economic sectors each ¹³.

Finally, they mention in line 395 that they use the World Bank GDP, but which one (nominal, real, in PPP, in Int\$)?

RE: Done.

All other socioeconomic data (for example, population and GDP in PPP (constant 2017 international \$)) used in this study were obtained from the World Bank ¹⁶.

The modelling of household consumption is rather simple. I understand that some simplifications are required when handling larger models. However, since the study concentrates on consumer choices, I think it would be relevant to consider both income effects and price effects, particularly to embed the substitutions and complementarities across different goods, which relates to the comment I made regarding the limitation of the modelling of the rebound effects. If the study focused on production, I would better understand the simplification in the modelling of the demand changes. In the context of this paper, however, further efforts would be welcomed. If they are not undertaken, at least a justification or discussion of these limitations should be added.

RE: Thank you for your valuable comments and suggestions. Our model accounts for both income and substitution effects to analyze household demand changes due to lifestyle shifts. Below, we address your specific concerns:

1) Substitution effect:

In modelling household demand changes resulting from lifestyle shifts, we incorporate substitutions and complementarities within the lifestyle scenarios. Specifically, we adjust household final consumption by reducing expenditure on targeted goods and services and increasing it for alternative goods and services, while also considering relative price differences between these products to enhance the accuracy of our estimates. For example, when modelling a shift to a vegetarian diet, we decrease meat consumption and simultaneously increase the consumption of plant-based

alternatives, taking into account the price difference between these products. Another example is the shift to working from home, which we assume reduces the need for mobility, while also leading to increased electricity and fuel consumption due to more time spent at home. We have revised the methods section to improve clarity:

Changes in household final consumption

Within our low-carbon expenditure measures, household final consumption was modified by both directly reducing final consumption (\mathbf{Y}^{red}) and increasing it due to the substitution effect (\mathbf{Y}^{sub}). The new household final consumption vectors can be calculated as:

$$\mathbf{Y}' = \mathbf{Y} - \mathbf{Y}^{red} + \mathbf{Y}^{sub} \quad \text{Equation 1}$$

Equation 2 shows the process for computing the reduced final demand (\mathbf{Y}^{red}).

$$\mathbf{Y}^{red} = \mathbf{Y} \odot (\mathbf{q}^y \times t) \quad \text{Equation 2}$$

where the column vector \mathbf{q}^y represents the household technical potential for final consumption of products from different countries, with values of elements ranging from 0 to 1. $\mathbf{q}^y \times t$ also reflects the disparity between the intended intervention and the actual adoption rate.

In terms of lifestyle change measures involving shifting effects, where substitution effects occur, the sum of all elements in the row vector \mathbf{Y}^{red} is substituted by other specific products. For example, when shifting to a vegetarian diet, plant-based foods, and dairy are substituted products. The increased final demand due to the substitution effect (\mathbf{Y}^{sub}) is distributed according to the current consumption patterns of specific population groups. For each household consumption vector, the element y_k^{sub} in \mathbf{Y}^{sub} of product k can be given by:

$$y_k^{sub} = \begin{cases} p \cdot \mathbf{1} \cdot \mathbf{Y}^{red} \cdot \frac{y_k}{\sum_k y_k}, & \text{if } k \text{ is a substituted product} \\ 0, & \text{otherwise} \end{cases} \quad \text{Equation 3}$$

where $\mathbf{1}$ is a column vector consisting of all 1s. The matrix multiplication $\mathbf{1} \cdot \mathbf{Y}^{red}$ allows us to obtain the summation of all elements in \mathbf{Y}^{red} . p is the relative price difference between the reduced products and substituted products. $\frac{y_k}{\sum_k y_k}$ denotes the proportion of final consumption accounted for by product k relative to the total final consumption of all substituted products.

2) Income effect:

When analyzing rebound effects, we use Engel curves to assess household re-spending patterns, focusing on how households allocate freed-up funds resulting from lifestyle changes. Specifically, we employed marginal expenditure shares (or marginal propensity to spend) to model the income effect on re-spending patterns of these savings. To better convey this approach, we have revised the methods section, providing clearer explanations of our modeling techniques and the rationale behind our use of Engel curves and marginal expenditure shares:

In our model, adopting low-carbon lifestyles can lead to money savings ($\Delta y = \mathbf{1} \cdot (\mathbf{Y} - \mathbf{Y}')$), which consumers would re-spend. This phenomenon, known as the rebound effect, involves reallocating saved money into final consumption, potentially leading to increased upstream carbon emissions throughout the supply chain¹⁷⁻¹⁹. Our study focuses on indirect rebound effects, where saved money is redirected to other goods and services^{20,21}. Direct rebound effects, involving increased consumption of the same fuel products due to energy efficiency gains, are considered unlikely in the context of low-carbon lifestyle changes^{18,21,22}. To model the impact of these changing budget constraints, we employed Engel curves, a

method widely used in the literature. While both marginal expenditure shares (or marginal propensity to spend) and income elasticities are commonly used, we chose to use marginal expenditure shares due to their methodological simplicity. This approach allows us to describe the additional expenditure on a specific commodity for every dollar increase in total expenditure, providing a clear and straightforward means of estimating the re-spending patterns associated with income effects.

3) Price effect:

We acknowledge that our current model does not explicitly account for price effects, which is a limitation when analysing counterfactual scenarios of lifestyle changes. This simplification was made to manage the complexity of the model and to focus on the broader implications of lifestyle changes on household consumption. We have now added this point to the limitations section of the paper to acknowledge the potential impact of omitting price effects on our results:

Additionally, we acknowledge that not explicitly incorporating price effects limits the model's ability to capture consumer behaviours fully²³.

In their conclusions, the authors comment on the challenges of promoting low-carbon lifestyle changes in economies attached to the growth paradigm poses a challenge, and then they mention the necessity of promoting frugal behaviors, especially among the higher-income groups. This analysis begs the question of how to achieve these changes. I understand this is beyond the scope of this paper. However, discussing possible policies to promote such lifestyle changes would be a great addition to the study. There is a vast literature of studies simulating the effects of policies promoting several of the lifestyle changes considered. The authors could discuss: (1) What are low-hanging fruits? (2) What is the evidence of different policies for the different goals? (3) Is (Are) there (some) a policy agenda(s) that would make more sense given their results? (4) Does this vary across countries or income groups?

RE: thanks for your suggestions. We have revised the discussion section according to your comments. please see the updated paragraphs below.

This research shows the potential of adopting low-carbon lifestyle changes among high-carbon households, leading to a considerable contribution to GHG emission mitigation at a global scale. Our estimations reveal that implementing a combination of low-carbon lifestyles in 29.7% of the top-emitting population could potentially result in a 39.3% reduction in global GHG emissions. We observed that nations in North America, Europe & Central Asia, and Latin America exhibit substantial potential for reducing GHG emissions mainly due to their high per-capita carbon footprints and the large number of households involved in our lifestyle-oriented mitigation scenarios. An interesting finding of this article is the unexpected demand-side mitigation possibilities observed in some countries in Sub-Saharan Africa, such as Mauritius, Namibia, and South Africa, which have been overlooked in previous studies.

When exploring emission hotspots among carbon-exceeding populations, we identified key consumption categories with high climate relevance such as buildings, food, and manufactured products. Within the buildings category, major contributors to emissions encompass home energy use, construction works, and building material use. Our mitigation scenarios highlight the large potential for carbon reductions through low-carbon lifestyle changes, particularly in services, diet, and buildings. Interventions such as reducing food waste, limiting long-distance travel and leisure activities, and shifting towards plant-based diets offer immediate and tangible benefits. These 'low-hanging fruits' represent key opportunities for policymakers to achieve progress with relatively straightforward measures. However, measures targeting the clothing sector, yield only little impact on GHG reductions primarily stemming from the inherently low-

carbon-intensive characteristics of the clothing sectors. Thus, while impactful and easily implementable interventions are available, effective emissions reduction requires prioritizing the most significant sources and opportunities for change.

The question of how to realize low-carbon lifestyle changes is pivotal, as an individual's choices are intricately linked to income and consumption levels, willingness, resource accessibility, and fiscal and policy frameworks^{6,24}. A combination of regulatory, economic, and information-based instruments, referred to as a “policy package”, is generally more effective in achieving these transitions than relying on single policy instruments alone²⁵. Governments worldwide have initiated various policies to support lifestyle changes, especially in critical areas of buildings, diet, and mobility. For example, countries such as Spain and Pakistan have promoted shorter working times and encouraged remote work to conserve energy⁵³. Investments in transport infrastructure, such as cycle lanes and high-speed rail systems, facilitate a shift towards more sustainable modes of mobility²⁷. In response to the energy price crisis triggered by the Russian-Ukrainian conflict, European nations like Germany, the Netherlands, and France, have initiated campaigns aimed at fostering energy-saving actions, including lowering the heating temperature and reducing showers^{15,26,28}. Furthermore, carbon pricing mechanisms, such as taxes and cap-and-trade systems, have proven effective in altering consumer behaviour by incorporating the environmental costs of carbon-intensive goods²⁹. Subsidies and tax incentives for renewable energy adoption, alongside information-based interventions such as carbon labelling and customized information feedback^{30–33}, have facilitated the transition towards greener consumption patterns. The effectiveness and feasibility of these policies are likely to vary across different countries and income groups. In developed nations, where infrastructure and resources are already in place, more aggressive measures may be viable³⁴. Similarly, low-income countries should avoid investing in carbon-intensive infrastructures that would lock them into high carbon-intensive expenditures³⁵. For high-income groups, policies could focus on curbing the consumption of luxury goods with high carbon footprints, potentially through progressive taxation or incentives for adopting low-carbon alternatives^{29,36,36}. Conversely, in developing countries and lower-income regions, the priority might be on improving access to affordable and sustainable options, ensuring that low-carbon lifestyle changes do not exacerbate existing inequalities^{8,37}.

Implementing low-carbon lifestyle changes comes with its own set of challenges, particularly the risk of unintended consequences such as rebound effects, where the cost savings from adopting low-carbon lifestyles could lead to increased consumption elsewhere, partially offsetting carbon reductions. Depending on the re-spending scenarios employed in this study, the estimated indirect rebound effects range from 5.9% to 40.4% of the expected global carbon reductions, potentially offsetting carbon savings between 0.8 Gt CO₂e and 5.2 Gt CO₂e. Addressing the rebound effects linked to lifestyle changes involves suggested pathways such as improving energy efficiency across various sectors, reducing consumption, and transitioning to greener consumption patterns^{19,38}. While energy efficiency improvements are effective, they are constrained by current technologies and require substantial investments. Downsizing carbon-intensive consumption appears more immediately accessible, particularly to wealthier populations due to their financial security and capacity to forgo non-essential goods and services. However, in the long term, aligning policies aimed at reducing demand with the existing GDP-based economic growth paradigm poses a challenge^{39,40}. This study emphasises promoting frugal behaviours among higher-income groups within the context of consumption sufficiency. Additionally, encouraging consumers to re-spend saved expenses on relatively low-carbon-footprint products could help alleviate rebound effects. Examples include purchasing environmentally friendlier electronic products and smart home devices and opting for sustainable tourism practices, such as low-carbon vacations or eco-tourism^{41,42}.

In summary, low-carbon lifestyles can play a pivotal role in short and medium-term climate mitigation efforts by reducing energy demand and overall consumption. These actions offer a swift and effective means of curbing climate change and carry fewer environmental risks compared to the implementation of technology-based measures⁴³. Recent events like the COVID-19 pandemic and the global energy crisis

triggered by the Russia-Ukraine conflict have demonstrated that rapid, widespread, and profound changes in lifestyles are possible with government and civil society coordination ^{26,44}. However, we have to recognize that such lifestyle changes primarily through reducing household demand could lead to rebound effects due to re-spending elsewhere. Policymakers should pay attention to mitigating such unintended consequences when designing carbon-saving policies. Additionally, achieving substantial, lasting emissions reductions requires both demand-side measures and supply-side technology solutions. They provide complementary solutions rather than one being superior to the other. Implementation of supply-side technologies faces technical challenges and may require significant timing and investment. In this case, demand-side solutions provide breathing space for the deployment of long-term technology-based reduction measures.

Specific comments:

Line 417 & 91: cite Vita et al.

RE: Done. Please see the revised manuscript.

Reviewer 3:

The impact of demand-side shifts on global carbon footprints is a critical yet understudied area. This paper contributes significantly by analyzing the potential global impact of lifestyle changes—a relatively novel focus in this field. Although the study does not capture all possible effects, it provides a valuable initial estimate. The results indicate that if the wealthiest 25% of the global population adopted certain lifestyle changes—such as adopting vegan diets, living in passive houses, and reducing car usage—the total carbon footprint could be reduced by a substantial 40%. These changes would undoubtedly have profound economic and societal ripple effects, but as a preliminary exploration, the study's approach is, I believe, justifiable.

One of the main concerns with the manuscript is its reliance on household expenditure surveys combined with Environmentally Extended Input-Output Analysis (EE-IOA). While this methodology aligns well with production-based emissions and is standard for estimating global consumption-based carbon footprints, it may not be entirely appropriate to use "of the shelf" for the purpose of this study. A well-known limitation of EE-IOA models is that they tend to overestimate the carbon footprint of wealthier households due to the simplistic assumption that spending in a category directly correlates with carbon emissions in that category, while in reality there is a large heterogeneity among different products and suppliers. This assumption disproportionately impacts those who spend more on luxury goods and services, potentially skewing results. This limitation, while often recognized in the literature as a standard caveat in the limitations section, becomes particularly problematic here because the study's findings hinge on expenditure data from wealthier demographics that might be inflated. Given the study's significant claims about potential emission reductions, this methodological weakness warrants further scrutiny.

Suggested Improvements:

To strengthen their argument, the authors should attempt to cross-verify their estimated impacts with alternative methodologies. For instance, comparing the estimated emissions reduction potential for housing against the total emissions from housing within a country could provide a reality check. Ideally, using physical data to do the same estimation would strengthen the results and enhance credibility. Although research in this area is limited, incorporating findings from studies like Girod de Haan (2010) and André et al. (2024) could provide additional benchmarks to assess the reasonableness of the claims.

Final thoughts:

While I recognize the potential impact of the findings presented in this manuscript, I have significant reservations about the robustness of the results due to the methodological limitations discussed. Previous similar studies, such as those by Vita and Chancel, have acknowledged similar issues as mere limitations. However, in the context of this paper, where such assumptions could substantially influence the primary conclusions, merely noting these as limitations may not suffice. Given the potential overestimation of carbon footprints for wealthier households, which are central to the study's claims, it is crucial to address these concerns more comprehensively. It may be necessary for the authors to explore alternative methodologies or provide more substantial validations for their current approach before considering this study for publication. I have a few other remarks and

suggestions, but this is my main concern.

RE: We appreciate your insightful comments and valuable suggestions.

First, EEIO analysis is the most widely used approach for carbon footprinting especially for global studies. Our paper aims to assess demand-side reduction measures. In this context, EEIO is the most suitable approach for this study to model the consumption pattern changes among detailed sectoral expenditure information and consider the entire global value chains with respective production patterns and energy mix for each country, globally.

Second, we fully agree that household expenditure surveys combined with EEIO Analysis may overestimate the carbon footprint of wealthier individuals. As you noted, this is a well-recognized limitation of this methodology. In the revised manuscript, we included this in the limitation part:

Another limitation of the model is its reliance on household expenditure surveys combined with EEMRIO Analysis, which may overestimate the carbon footprint of wealthy individuals and consequently overestimate potential reductions⁴⁵. This is due to the simplistic assumption that spending in a category directly correlates with carbon emissions, while in reality, there is considerable heterogeneity among products and suppliers within each category. To address this, we conducted a verification analysis, detailed in the Supplementary Information.

Regarding cross-verifying the estimated reduction potentials, we carefully considered your suggestions. We reviewed the two papers you mentioned, as well as other related works. While these studies offer valuable insights, data limitations and differing research focuses prevented a direct comparison. Girod and De Haan (2010) used a household consumption model based on functional units (e.g., kg of food, square meters of living space) to assess how rising affluence impacts consumption and to compare GHG emissions allocation by monetary versus functional units. Their study focused on Switzerland from 2002 to 2005, and connected data from the Swiss expenditure survey in physical units to Life Cycle Assessment (LCA) processes to calculate potential GHG reductions. This LCA approach converted data in physical units into GHG emissions per monetary unit, enabling a comparison between the two allocation methods (monetary versus functional). Although their methodology would be useful for validating, we were unable to obtain survey-based expenditure data in physical units for our study period, particularly from data sources consistent with our current model.

To address this concern, we conducted a sensitivity analysis comparing our household-specific EEMRIO model with national-level input-output tables that do not distinguish between expenditures for different household categories/expenditure groups and therefore 'average out' the differences in prices and thus carbon emissions for specific consumption items. The national-level tables provide an aggregation of each country's household final consumption into a single vector. While the national-level model provides a less detailed view of consumption patterns, it serves as a useful benchmark as it provides the overall emissions for a consumption item for each country irrespective of price and quality differences which are the limitations of inequality studies based on monetary survey data. The results indicate that the differences in emission reduction potentials between the two models almost all fall within a 5% range, suggesting that our global MRIO linked to consumer

expenditure survey provides a reliable estimate of emission reduction potentials. More details are provided below.

Supplementary Information: Limitations and Verification of Results

A key limitation of this study lies in the reliance on household expenditure surveys combined with Environmentally Extended Multi-Regional Input-Output (EEMRIO) Analysis. This methodology may overestimate the carbon footprint of wealthy individuals due to the assumption that spending directly correlates with emissions. In reality, there is substantial heterogeneity in the carbon intensity of products and services within each expenditure category. For instance, a rich individual's expenditure on a product might be disproportionately reflected in their carbon footprint compared to a low-income individual purchasing a less expensive but functionally similar product. This could potentially lead to an overestimation of the carbon emissions of an item for the luxury version of the same product that would be bought for a much higher price and thus could inflate the estimated potential for GHG emissions reductions in our analysis.

To address this, we conducted a sensitivity analysis by comparing our household-specific MRIO database with national-level MRIO tables with aggregated final household demand. The national-level model aggregated each country's household final consumption into a single vector, assuming a homogenous population with identical consumption patterns, implying that all households within a country have the same average carbon footprint. We then applied the same set of low-carbon lifestyle measures to both models to calculate the national-level reduction potential. For the household-specific model, we aggregated household-level emission reductions to derive the national-level reduction potential. In the national-level MRIO model, we assumed that the same proportion of the population adopted identical low-carbon lifestyle measures and calculated the corresponding national-level reduction potential. Comparing these national-level results allows us to assess the impact of differentiating households based on expenditure levels.

As illustrated in Supplementary Figure 6, the differences in emissions reduction potential between the two models range from -2.6% to 23.1% across various countries and lifestyle measures, with 97% of these differences being less than 5%. This indicates a relatively high degree of consistency between the models, suggesting that our household-specific model provides a reliable estimate of emission reduction potentials. While the model may slightly overestimate the carbon footprints of high-consumption individuals, particularly in the context of diet-related measures, the overall trends and patterns identified in the study are robust. This overestimation of diet-related measures is likely attributable to the consumption of processed foods, imported products, and luxury food items by wealthier people.

Supplementary Figure 1. Differences in emissions reduction potentials across countries and low-carbon expenditures. The Y-axis shows reduction potentials between the household-specific MRIO database with national-level MRIO tables with aggregated final household demand.

In terms of Andre et al (2024), it provided global data on individuals' willingness to mitigate climate change, finding widespread support for climate action worldwide, which aligns with our study's narrative. Our study designed low-carbon lifestyle measures for high-emitting households globally but did not incorporate their willingness to adopt these measures. While this aspect is beyond our current scope, we have included it in the discussion and treated it as a potential area for future research. We appreciate your suggestion and will consider it in future work.

The question of how to realize low-carbon lifestyle changes is pivotal, as an individual's choices are intricately linked to income and consumption levels, willingness, resource accessibility, and fiscal and policy frameworks^{6,24}. A combination of regulatory, economic, and information-based instruments, referred to as a "policy package", is generally more effective in achieving these transitions than relying on single policy instruments alone²⁵.

References

1. van den Berg, N. J. *et al.* Improved modelling of lifestyle changes in Integrated Assessment Models: Cross-disciplinary insights from methodologies and theories. *Energy Strategy Reviews* **26**, 100420 (2019).
2. Saujot, M., Le Gallic, T. & Waisman, H. Lifestyle changes in mitigation pathways: policy and scientific insights. *Environ. Res. Lett.* **16**, 015005 (2020).
3. Duchin, F. & Hubacek, K. Linking social expenditures to household lifestyles. *Futures* **35**, 61–74 (2003).
4. Baiocchi, G., Feng, K., Hubacek, K. & Walters, C. Carbon footprint of American lifestyles: a geodemographic segmentation approach. *Environ. Res. Lett.* **17**, 064018 (2022).
5. Vita, G. *et al.* The Environmental Impact of Green Consumption and Sufficiency Lifestyles Scenarios in Europe: Connecting Local Sustainability Visions to Global Consequences. *Ecological Economics* **164**, 106322 (2019).
6. United Nations Environment Programme. *Emissions Gap Report 2020*. <https://www.unep.org/emissions-gap-report-2020> (2020).
7. McCauley, D. & Heffron, R. Just transition: Integrating climate, energy and environmental justice. *Energy Policy* **119**, 1–7 (2018).
8. Chancel, L. Global carbon inequality over 1990–2019. *Nat Sustain* 1–8 (2022) doi:10.1038/s41893-022-00955-z.
9. Bruckner, B., Hubacek, K., Shan, Y., Zhong, H. & Feng, K. Impacts of poverty alleviation on national and global carbon emissions. *Nat Sustain* **5**, 311–320 (2022).
10. Hubacek, K., Baiocchi, G., Feng, K. & Patwardhan, A. Poverty eradication in a carbon constrained world. *Nat Commun* **8**, 912 (2017).
11. *Emissions Trends and Drivers*. In *IPCC, 2022: Climate Change 2022: Mitigation of Climate Change. Contribution of Working Group III to the Sixth Assessment Report of the Intergovernmental Panel on Climate Change*. 215–294 https://www.cambridge.org/core/product/identifier/9781009157926%23c2/type/book_part (2023) doi:10.1017/9781009157926.004.
12. The World Bank. Global Consumption Database. <https://datatopics.worldbank.org/consumption/> (2024).
13. Aguiar, A., Chepeliev, M., Corong, E. & Mensbrugghe, D. van der. The Global Trade Analysis Project (GTAP) Data Base: Version 11. *Journal of Global Economic Analysis* **7**, (2022).
14. IEA. World Energy Balances. <https://www.iea.org/data-and-statistics/data-product/world-energy-balances> (2023).
15. Guan, Y. *et al.* Burden of the global energy price crisis on households. *Nat Energy* **8**, 304–316 (2023).
16. The World Bank. World Development Indicators. <https://data.worldbank.org/> (2023).

17. Haque, M. O. *Income Elasticity and Economic Development: Methods and Applications*. (Springer Science & Business Media, 2005).
18. Druckman, A., Chitnis, M., Sorrell, S. & Jackson, T. Missing carbon reductions? Exploring rebound and backfire effects in UK households. *Energy Policy* **39**, 3572–3581 (2011).
19. Font Vivanco, D., Kemp, R. & van der Voet, E. How to deal with the rebound effect? A policy-oriented approach. *Energy Policy* **94**, 114–125 (2016).
20. Grabs, J. The rebound effects of switching to vegetarianism. A microeconomic analysis of Swedish consumption behavior. *Ecological Economics* **116**, 270–279 (2015).
21. Font Vivanco, D., Freire-González, J., Kemp, R. & van der Voet, E. The Remarkable Environmental Rebound Effect of Electric Cars: A Microeconomic Approach. *Environ. Sci. Technol.* **48**, 12063–12072 (2014).
22. Zha, D., Chen, Q. & Wang, L. Exploring carbon rebound effects in Chinese households' consumption: A simulation analysis based on a multi-regional input–output framework. *Applied Energy* **313**, 118847 (2022).
23. Wollburg, P., Hallegatte, S. & Mahler, D. G. Ending extreme poverty has a negligible impact on global greenhouse gas emissions. *Nature* **623**, 982–986 (2023).
24. Andre, P., Boneva, T., Chopra, F. & Falk, A. Globally representative evidence on the actual and perceived support for climate action. *Nat. Clim. Chang.* 1–7 (2024) doi:10.1038/s41558-024-01925-3.
25. van Sluisveld, M. A. E., Martínez, S. H., Daioglou, V. & van Vuuren, D. P. Exploring the implications of lifestyle change in 2 °C mitigation scenarios using the IMAGE integrated assessment model. *Technological Forecasting and Social Change* **102**, 309–319 (2016).
26. IEA. *Playing My Part-How to Save Money, Reduce Reliance on Russian Energy, Support Ukraine and Help the Planet*. <https://www.iea.org/reports/playing-my-part> (2022).
27. International Transport Forum. *Long-Term Policies Yield Significant Modal Shifts in Investment*. <https://www.itf-oecd.org/modal-shift-cleaner-transport-fails-materialise> (2023).
28. Zhang, Y. *et al.* Energy price shocks induced by the Russia-Ukraine conflict jeopardize wellbeing. *Energy Policy* **182**, 113743 (2023).
29. Oswald, Y., Millward-Hopkins, J., Steinberger, J. K., Owen, A. & Ivanova, D. Luxury-focused carbon taxation improves fairness of climate policy. *One Earth* **0**, (2023).
30. Khosla, R., Sircar, N. & Bhardwaj, A. Energy demand transitions and climate mitigation in low-income urban households in India. *Environ. Res. Lett.* **14**, 095008 (2019).
31. Taufique, K. M. R. *et al.* Revisiting the promise of carbon labelling. *Nat. Clim. Chang.* **12**, 132–140 (2022).
32. Khanna, T. M. *et al.* A multi-country meta-analysis on the role of behavioural change in reducing energy consumption and CO2 emissions in residential buildings. *Nat Energy* **6**, 925–932 (2021).
33. Nemati, M. & Penn, J. The impact of information-based interventions on conservation behavior: A

- meta-analysis. *Resource and Energy Economics* **62**, 101201 (2020).
34. *Demand, Services and Social Aspects of Mitigation*. 503–612
https://www.cambridge.org/core/product/identifier/9781009157926%23c5/type/book_part
 (2023) doi:10.1017/9781009157926.007.
 35. Kobayakawa, T. The carbon footprint of capital formation: An empirical analysis on its relationship with a country's income growth. *Journal of Industrial Ecology* **26**, 522–535 (2022).
 36. Institute for Global Environmental Strategies, Aalto University & D-mat Ltd. *1.5-Degree Lifestyles: Targets and Options for Reducing Lifestyle Carbon Footprints*. (2019).
 37. Baltruszewicz, M. *et al.* Household final energy footprints in Nepal, Vietnam and Zambia: composition, inequality and links to well-being. *Environ. Res. Lett.* **16**, 025011 (2021).
 38. Font Vivanco, D. *et al.* Rebound effect and sustainability science: A review. *Journal of Industrial Ecology* **26**, 1543–1563 (2022).
 39. Hickel, J. *et al.* Degrowth can work — here's how science can help. *Nature* **612**, 400–403 (2022).
 40. Hickel, J. & Kallis, G. Is Green Growth Possible? *New Political Economy* **25**, 469–486 (2020).
 41. Schill, M., Godefroit-Winkel, D., Diallo, M. F. & Barbarossa, C. Consumers' intentions to purchase smart home objects: Do environmental issues matter? *Ecological Economics* **161**, 176–185 (2019).
 42. Sovacool, B. K., Newell, P., Carley, S. & Fanzo, J. Equity, technological innovation and sustainable behaviour in a low-carbon future. *Nat Hum Behav* **6**, 326–337 (2022).
 43. Creutzig, F. *et al.* Towards demand-side solutions for mitigating climate change. *Nature Clim Change* **8**, 260–263 (2018).
 44. Kikstra, J. S. *et al.* Climate mitigation scenarios with persistent COVID-19-related energy demand changes. *Nat Energy* **6**, 1114–1123 (2021).
 45. Girod, B. & De Haan, P. More or Better? A Model for Changes in Household Greenhouse Gas Emissions due to Higher Income. *Journal of Industrial Ecology* **14**, 31–49 (2010).

Response letter to Reviewers' comments

Reviewer 2:

I would like to thank the authors for taking the time to thoroughly review the paper and seriously consider my suggestions. I think both the inclusion of all GHG and the rewriting of the discussion including possible policy agendas improved substantially the quality of the paper. I would also like to thank you for clarifying my misunderstandings and for making the text clearer for other readers. Regarding the reviewed version, I have only a couple of comments left.

First, in line 524 I think you mean that when switching to a vegetarian diet, plant-based foods, and dairy products (and probably also eggs) are substitutes. Using "substituted products" is confusing because this means that they are substituted away. I think you mean that dairy and plant-based products can be used instead of meat in which case the correct word is substitute, as the very pertinent example in the Cambridge dictionary indicates: Tofu can be used as a meat substitute in vegetarian recipes (<https://dictionary.cambridge.org/dictionary/english/substitute>).

RE: Done. Thank you for the suggestion.

For example, when shifting to a vegetarian diet, plant-based food, dairy, and eggs are substitutes.

Second, I have only one remaining comment regarding the methodology. I don't necessarily think it should lead to a change in the analysis but to an acknowledgment of a limitation and perhaps potential future improvements. In lines 285-287 you express that the freed-up funds are distributed proportionally (based on their marginal propensity to spend) among the elements of the same category and other goods. The limitation I see in this approach is that the marginal propensity to spend is calculated under current consumption and the y^{sub} is also distributed according to the current consumption. The two elements affecting this substitution are the relative price difference between the two goods and the proportion of final consumption accounted for by substitute k . You are, however, modeling a change in consumption that in reality goes beyond differences in relative prices and the current consumption share of a good. Say for example that I go vegetarian but before I already ate a large portion of legumes in my diet, since dairy is relatively cheap with respect to meat and I might perceive it more as a closer substitute to meat than legumes, I might substitute almost all the consumption of meat by dairy products and keep my legumes consumption almost constant. This might be a serious concern because if I previously ate mainly poultry and fish, substituting them for dairy products might not lead to the expected emissions reductions. Such consumer preferences are normally modeled by own-price and cross-price elasticities. The insights you can draw from your demand system are constrained by the fact that you do not include such demand elasticities (also called preference parameters sometimes). Therefore I think that this proportionality based on the marginal propensity to spend is a rather limited way of modelling substitution effects in consumption.

I am aware of what you wrote in line 613, I however do not think it is enough because own-price and cross-price elasticities depend on various socioeconomic aspects, reflecting different price effects on different economies (see Bouyssou, C. G., Jensen, J. D. & Yu, W. Food for thought: A meta-analysis of animal food demand elasticities across world regions. *Food Policy* 122, (2024)). The fact that price effects vary across world regions and income groups is therefore also missing from your analysis and could potentially lead to significantly different results.

In case the authors would like to have a reference of a (also simple) demand system using own- and cross-price elasticities, I think a good example is: Robinson, S. et al. The International Model for Policy Analysis of Agricultural Commodities and Trade (IMPACT): Model Description for Version 3. *SSRN Journal* (2015) doi:10.2139/ssrn.2741234. That, for example, has been used to model consumption changes stemming from emissions pricing: Springmann, M. et al. Mitigation potential and global health impacts from emissions pricing of food commodities. *Nature Clim Change* 7, 69–74 (2017). Other more complicated demand systems considering such preference parameters are included in the GTAP and other general and partial computable general equilibrium models as is customary in the field of Economics.

For a discussion on the role of preference parameters/demand elasticities in demand systems see:

1. Valin, H. et al. The future of food demand: understanding differences in global economic models. *Agricultural Economics* 45, 51–67 (2014).
2. Ho, M. et al. Modelling Consumption and Constructing Long-Term Baselines in Final Demand. *Journal of Global Economic Analysis* 5, 63–108 (2020).

Summing up, I think the efforts made by the authors are valuable and shed new light on crucial aspects. I would like to recommend it for publication. Nevertheless, I think it is very important to acknowledge the aforementioned limitations on modeling consumer behavior without using preference parameters as they could significantly influence the results and key messages of the paper if they were included.

RE: We sincerely appreciate the reviewer's thoughtful comments on our methodological choices.

Our analysis does not include own-price and cross-price elasticities¹⁻³. While we acknowledge that relying on marginal propensities to spend to allocate freed-up incomes has limitations, this approach offers a straightforward yet meaningful way to estimate potential emission reductions. The primary aim of our study is to explore the carbon savings that could result from a hypothetical shift to low-carbon expenditures. It is designed as a "what-if" scenario to examine the redistribution of freed-up funds among households with different consumption patterns. Therefore, our focus is not on capturing consumer behaviour in response to price changes or policy nudges, as these aspects are beyond the scope of our analysis. We have now explicitly noted this limitation in the methods section of the manuscript. Once again, thank you for your thoughtful feedback and for recommending relevant literature to improve the rigour and clarity of our work.

Additionally, we acknowledge that not explicitly incorporating price changes and policy nudges limits the model's ability to fully capture dynamic consumer behaviours¹⁻³. However, modeling such behaviors lies beyond the scope of our current study, which focuses on estimating potential carbon savings under hypothetical redistribution scenarios.

In modelling the substitution effects (i.e., the distribution of Y^{sub}), such as when transitioning to a vegetarian diet, plant-based food, dairy, and eggs serve as substitutes. In the previous version of our analysis, Y^{sub} was allocated based on current consumption patterns. Inspired by your feedback, we have revised this approach. The updated approach allocates Y^{sub} according to the marginal propensity to spend ^{4,5}. Using the vegetarian example again, the updated approach redistributes savings from reduced meat consumption among substitutes (plant-based foods, dairy, and eggs) based on how likely consumers will increase spending on these substitutes in response to higher incomes, rather than simply reflecting their current consumption preferences. As a result, we have restructured the methods section and updated the results throughout the manuscript. Please refer to the revised manuscript.

Reviewer 3:

My main issue with this manuscript remains the potential inflation of results due to the method used. The authors have acknowledged the limitations of the EE-IOA framework and provided a sensitivity analysis, as requested. While I appreciate their efforts, I am not entirely convinced, and I would like to see further clarification in their response.

The EE-IOA framework assumes a linear relationship between spending and carbon footprints, which is reasonable at the aggregate level. However, this assumption becomes problematic when applied to sub-groups that differ significantly from the average, such as high-income households, as examined in this manuscript. These households typically spend more on goods and services, but this does not necessarily reflect their “true” carbon footprint. For example, purchasing a €1000 suit does not generate 100 times the emissions of a €10 T-shirt. The same applies to major investments like housing, where location heavily influences market value, or vehicles, where price often reflects non-emission-related factors. Food and services also show price premiums for elements like branding or expertise that don't correlate with higher emissions. This indicates both price and product heterogeneity that the EE-IOA framework fails to capture. This limitation is widely acknowledged in studies using the EE-IOA framework and, addressing it in this manuscript could, I believe, contribute to advancing the literature on carbon footprint estimations.

In my first review and as a first point, I suggested that the authors should try to cross-verify their national-level reduction potentials with primary data, such as comparing estimated savings from dietary shifts with FAOSTAT or household energy using EUROSTAT emissions data. While I understand that this would introduce other uncertainties and may be a demanding request, I believe such cross-verification would add valuable validation. The Andre et al. (2024) paper I referenced can be found here: <https://www.sciencedirect.com/science/article/pii/S0921800924000193>.

Requested action: I leave it to the authors to decide how to proceed with this.

RE: Thank you for the suggestions. We conducted two case studies—Switzerland and China—using physical consumption data to cross-verify carbon footprinting by linking MRIO data with household budget survey data. Additionally, we assessed reduction potentials under various low-carbon expenditure measures. Please refer to the Supporting Information file and the detailed analysis below:

Cross-verification: physical units vs. monetary units

To address uncertainties arising from price heterogeneity within sub-population groups, we cross-validated carbon footprint calculations using physical consumption data (e.g., quantities consumed)^{6–8}. This approach was applied to household-level data from Switzerland and China, allowing us to assess carbon footprints and reduction potentials under illustrative low-carbon expenditure scenarios.

Switzerland case study

Switzerland, one of the wealthiest countries, provides highly detailed household expenditure data via its Household Budget Survey (HBS)⁸. The HBS dataset for 2015–2017 includes information on both physical consumption and monetary expenditure data for food across five income groups (monthly income ranging from 0 to 12,856 Swiss francs)⁹. We linked this data with Switzerland's household final demand data from

the GTAP MRIO data to calculate food-related carbon footprints for each income group using both in physical and monetary units.

Supplementary Fig. 7. Switzerland food-related carbon footprints and reduction potential from two low-carbon expenditures across five income groups in 2017. The plot on the right compares the reduction potentials derived from monetary-based and physical-based calculations.

Supplementary Fig. 7 compares the carbon footprint and reduction potential derived from monetary-based and physical-based calculations. Carbon footprints calculated with monetary data ranged from 1.35–1.56 tons CO₂e/cap/year, whereas physical data produced slightly narrower estimates of 1.44–1.54 tons CO₂e/cap/year. Monetary data tend to slightly overestimate footprints for wealthier households and underestimate them for lower-income households compared to physical data. However, the overall differences are small, with percentage deviations (relative to physical data-based results) ranging from -6.0% to 11.0%.

Under two low-carbon food expenditure scenarios—1) adopting a healthy vegan diet and 2) reducing food waste—both calculation methods produced highly consistent carbon savings estimates, reinforcing the robustness of our approach.

China Case Study

China, a rapidly developing country with considerable variation in population expenditure levels, was selected as a case study. We utilized the 2014 Urban Household Income and Expenditure Survey (UHIES)¹⁰, the latest dataset available. Based on stratified multistage random sampling, the 2014 UHIES dataset covers over 13,000 urban households across four provinces (Liaoning, Sichuan, Guangdong, and Shanghai). The dataset provides detailed household-level data on demographics (e.g., size, income, dwelling characteristics) and expenditures for 145 products, 59 of which include physical consumption data (e.g., food, electricity, and private cars). For products lacking physical data, such as school accommodation fees, monetary expenditures were used as proxies for physical consumption, as is commonly used in the literature^{8,10}.

We processed the micro-level data and grouped households into deciles based on their expenditure levels. These subgroup consumption patterns were then bridged with China’s household final demand data from the GTAP MRIO data to calculate carbon footprints for each expenditure group in physical and monetary units.

Supplementary Fig.8. Carbon footprints and reduction potential from low-carbon expenditures across population deciles of Chinese households in 2017. The plot below compares the reduction potential derived from monetary-based and physical-based calculations.

As shown in Supplementary Fig.8, Carbon footprints using monetary data ranged from 1.0 to 11.9 tons CO₂e/cap/year, while physical data produced similar estimates of 1.1 to 11.7 tons CO₂e/cap/year ranging from -7.6% to 1.9%. As in the Switzerland case study, monetary data slightly overestimates footprints for higher-income households and underestimates them for lower-income households. Despite these small discrepancies the overall trend and pattern remain consistent. By comparing results using monetary data from the World Bank Global Consumption Dataset (WBGCD) in the main text (0.7 to 11.7 tons CO₂e/cap/yr) with those derived from the UHIES data, we conclude the approach used in this study provides robust estimates of carbon footprints.

We analysed seven low-carbon expenditure scenarios for carbon-exceeding households (the same population proportions as in the main text) including areas in Buildings (Passive House), Clothing (Durable Fashion), Diet (Healthy Vegan), Food (Less Waste), Manufactured Products (Share & Repair), Mobility (Fewer Cars), and Services (Nonmarket Services). The results were consistent across both physical and monetary units, demonstrating the robustness of our methodology.

The authors propose a sensitivity analysis in the rebuttal, comparing their estimated emission reductions to those obtained if the same changes were made by the corresponding proportion of "average consumers" in each country using national-level MRIO models. Since the average consumer

represents the average household emissions, they argue that significant deviations for high-income households would highlight heterogeneity. They find small deviations and conclude that this should not be an issue. However, I'm not convinced this fully addresses the issue. Even if the relative reduction potentials for both groups appear similar in many cases, this doesn't seem to me to resolve the concern about inflated emissions among high-income households, since it just reflects the relative reduction potential of the already potentially inflated numbers.

RE: Thank you for pointing that out. We agree that the comparison of reduction potential only reflects the relative reductions based on the already higher carbon footprint estimates for rich households. To address this, we have compared both the carbon footprint and the reduction potential, as per your earlier comment. Meanwhile, we have retained the sensitivity analysis and treated it as a benchmark in the supporting information, as suggested in your subsequent comments.

Requested action 1: I would appreciate an illustrative example of one or two of these calculations and I think they could also help clarify this section.

RE: Done. Please see our response to your previous comment.

Requested action 2: Additionally, I suggest the authors present the total summarized effect of all national-level reduction potential estimated for the average consumers and include it in the article. This estimate would serve as a benchmark estimate for the effect of the suggested changes, were they to be adopted randomly by the same share of households in each country.

RE: Thank you for the suggestion. We have added a comparison in the main text. Please refer to the paragraph and Figure 6 below:

Figure 6 illustrates significant deviations (ranging from -7% to 33%) between carbon savings achieved by targeting carbon-exceeding households with a combination of low-carbon expenditure measures and those obtained by applying the same changes to a corresponding proportion of "average consumers" in each country—a common practice in the literature^{6,7}. These deviations highlight the heterogeneity in household consumption patterns, with high-emitting households demonstrating a higher potential for reducing their carbon footprint. Notably, we found that some Sub-Saharan African countries, such as Mauritius (achieving a 52.5% reduction in this study), Namibia (45.6%), and Chad (44.7%), often overlooked in previous studies, display relatively higher deviations. Namibia's reductions are driven by changes in food, mobility and services expenditures, due to its heavy dependence on the tourism industry.

Figure 6. Comparison of national carbon savings through low-carbon expenditures implemented by carbon-exceeding households and randomly selected average consumers. The y-axis represents national carbon savings achieved by carbon-exceeding households through low-carbon expenditures. The x-axis shows the reductions resulting from applying the same changes to an equivalent proportion of “average consumers” in each country.

Note: The assumption of 27% reduction from “less waste” seems to have been confused with the measure “Food sufficiency”. The reference estimates the effect of less waste at 12% from what I understood.

Requested action: Look into it.

RE: Thanks for pointing out that. After reviewing the literature, we have determined that a 12% reduction in food waste is a more appropriate assumption for this study ^{11,12}. Accordingly, we have updated the parameter in the revised model and manuscript to reflect a 12% reduction in food consumption.

Comments with no requested actions:

As for modeling alternative re-spending scenarios and specifically the basis for SC3 this empirical article may be interesting: <https://www.sciencedirect.com/science/article/pii/S0959652622053136>

RE: Thanks for the suggestion. The paper you recommended models a scenario in which “all possible savings from a green behaviour or low-carbon consumption behaviour are instead spent on other goods and services at the marginal GHG intensity, estimated from cross-sectional comparisons of consumers” ⁵. This re-spending scenario is included in our current study.

1) To model the rebound effect of various low-carbon expenditure behaviours in our study, we use the same marginal redirected spending assumption (MRA).

Our initial evaluation focuses on rebound effects across 21 distinct low-carbon expenditure measures, where we assumed that all saved money is re-spent on products unaffected by specific low-carbon lifestyle changes, following consumers’ marginal expenditure patterns. For example, reduced air travel decreased spending on airfare but left expenditures on other non-aviation goods constant. The freed-up funds from

air travel savings were then distributed proportionally (based on their marginal propensity to spend) among these non-aviation travel items and other goods. We found considerable disparities in rebound effects across different lifestyles (see Supplementary Fig.4). Service-related expenditures tend to have greater backfire effects, as savings from cutting back on leisure activities are often redirected to areas with higher carbon intensity, such as increased spending on food or extended time at home (raising residential energy use).

2) When assessing the rebound effect of a combination of low-carbon expenditure behaviours, we design a similar re-spending scenario (SC1).

When exploring the potential magnitude of rebound effects resulting from combined lifestyle changes, we designed various alternative re-spending patterns, taking inspiration from refs ¹³⁻¹⁶. As shown in Fig 8, our analysis involves three rebound scenarios (SC1-SC3). SC1 assumes that all saved money is re-spent on products unaffected by low-carbon expenditures ¹¹. In this case, it is estimated that 4.8 Gt CO₂e GHG reductions are offset, resulting in a rebound effect of 45.8%. Re-spending on the buildings category leads to a reduction loss of 2.1 Gt CO₂e.

References

1. Springmann, M. *et al.* Mitigation potential and global health impacts from emissions pricing of food commodities. *Nature Clim Change* **7**, 69–74 (2017).
2. Bouyssou, C. G., Jensen, J. D. & Yu, W. Food for thought: A meta-analysis of animal food demand elasticities across world regions. *Food Policy* **122**, 102581 (2024).
3. Ho, M. *et al.* Modelling Consumption and Constructing Long-Term Baselines in Final Demand. *Journal of Global Economic Analysis* **5**, 63–108 (2020).
4. Lekve Bjelle, E., Steen-Olsen, K. & Wood, R. Climate change mitigation potential of Norwegian households and the rebound effect. *Journal of Cleaner Production* **172**, 208–217 (2018).
5. Andersson, D. & Nässén, J. Measuring the direct and indirect effects of low-carbon lifestyles using financial transactions. *Journal of Cleaner Production* **386**, 135739 (2023).
6. Kilian, L., Owen, A., Newing, A. & Ivanova, D. Microdata selection for estimating household consumption-based emissions. *Economic Systems Research* **35**, 325–353 (2023).
7. André, M., Bourgeois, A., Combet, E., Lequien, M. & Pottier, A. Challenges in measuring the distribution of carbon footprints: The role of product and price heterogeneity. *Ecological Economics* **220**, 108122 (2024).
8. Girod, B. & De Haan, P. More or Better? A Model for Changes in Household Greenhouse Gas Emissions due to Higher Income. *Journal of Industrial Ecology* **14**, 31–49 (2010).
9. Switzerland Federal Statistical Office. Switzerland Household Budget Survey, 2015-2017. <https://www.bfs.admin.ch/bfs/en/home/statistiken/wirtschaftliche-soziale-situation-bevoelkerung/einkommen-verbrauch-vermoegen/haushaltsbudget.html>.
10. Zhang, Y., Wang, F. & Zhang, B. The impacts of household structure transitions on household carbon emissions in China. *Ecological Economics* **206**, 107734 (2023).
11. Wood, R. *et al.* Prioritizing Consumption-Based Carbon Policy Based on the Evaluation of Mitigation Potential Using Input-Output Methods: Prioritizing Consumption-Based Carbon Policies. *Journal of Industrial Ecology* **22**, 540–552 (2018).
12. Vita, G. *et al.* The Environmental Impact of Green Consumption and Sufficiency Lifestyles Scenarios in Europe: Connecting Local Sustainability Visions to Global Consequences. *Ecological Economics* **164**, 106322 (2019).
13. Zha, D., Chen, Q. & Wang, L. Exploring carbon rebound effects in Chinese households' consumption: A simulation analysis based on a multi-regional input–output framework. *Applied Energy* **313**, 118847 (2022).
14. Grabs, J. The rebound effects of switching to vegetarianism. A microeconomic analysis of

Swedish consumption behavior. *Ecological Economics* **116**, 270–279 (2015).

15. Meshulam, T., Font-Vivanco, D., Blass, V. & Makov, T. Sharing economy rebound: The case of peer-to-peer sharing of food waste. *Journal of Industrial Ecology* **27**, 882–895 (2023).
16. Druckman, A., Chitnis, M., Sorrell, S. & Jackson, T. Missing carbon reductions? Exploring rebound and backfire effects in UK households. *Energy Policy* **39**, 3572–3581 (2011).

Response letter to Reviewers' comments

Reviewer 2:

I would like to thank the authors for their replies to my previous comments. I appreciate the change in the modeling of Y_{sub} , which I think is now more realistic.

In the first reply to my comment on the GHG emissions (first round of revisions) the authors wrote: After recalculating, the global production-based GHG emissions for 160 countries and regions in 2017 amount to 50.56 Gt CO₂e. The global GHG footprints of the 116 countries amount to 32.6 Gt CO₂e in 2017. The total reduction potential from a combination of low-carbon lifestyle measures is 12.8 Gt CO₂e, representing 39.3% of global GHG emissions in 2017.

Given these calculations, I think it is quite misleading to write (in lines 19-20 and 281-282) that: “a combination of low-carbon expenditures among the top 23.7% emitters could cut global carbon footprints by 10.4 Gt CO₂e (i.e., 40.1% of global GHG emissions in 2017)”. If global emissions are 50.56 Gt CO₂e in 2017, then the share of global emissions is 20.6%.

Else, the statement needs to be more precise, e.g., “a combination of low-carbon expenditures among the top 23.7% emitters could cut global carbon footprints by 10.4 Gt CO₂e (i.e., 40.1% of the GHG footprints of the 116 countries we consider in our study in 2017)”.

Also, taking the 32.6 Gt CO₂e in 2017 the authors used before (in the first reply to my comments) and the new estimate of 10.4 Gt CO₂e the percentage becomes 31.9%, not 40.1%. Is there a change in the GHG footprints estimates of the 116 countries you used since the last revision?

I think there is no need to inflate the results by using a different base without being transparent about the base used. A 20.6% of global emissions reduction is a significant number, already, and more accurate and transparent for the reasons I already explained. I hope the authors will understand my concerns about misrepresenting mitigation potentials.

RE: Thank you for raising these important concerns regarding the consistency and transparency of our calculations and interpretations. We acknowledge that there were some unclear and potentially misleading statements in our previous response, and we appreciate the opportunity to clarify and improve our explanations.

First, we use the consumption-based emissions of 116 countries as the benchmark for both global and national analyses, as our reduction potential analysis is targeted at households in these 116 countries. The production-based emissions (50.56 Gt CO₂e in 2017 across 160 countries and regions, as reported in the GTAP MRIO database) were used to calculate the carbon coefficient ϵ (i.e., carbon emissions per unit of economic output for all sectors in all countries) in equation (1), which serves as a further input for the EEMRIO model. As suggested, we have provided a clear explanation of this choice in the manuscript, explicitly clarifying the distinction between the two emission accounting approaches used:

We found that implementing a combination of low-carbon expenditures among the top 23.7% emitters could cut global carbon footprints by 10.4 Gt CO₂e (i.e., 40.1% of the consumption-based emissions of the 116 countries analysed in this study or 31.7% of the global total in 2017).

Second, regarding the carbon footprint calculation, we confirm that in this revision, we updated the approach used to bridge the household final demand (GTAP MRIO dataset) with the household expenditure data based on the World Bank's Global Consumption Database (WBGCD). The previous approach entirely depends on the expenditure structure from the household expenditure survey ^{1,2}. We adopted the updated approach because the GTAP dataset for the year 2017 is more recent compared to the WBGCD data for 2011. After the recalculation, the GHG footprints (i.e., consumption-based GHG emissions) of the 116 countries amounted to 25.9 Gt CO₂e in 2017. The total reduction potential from a combination of low-carbon lifestyle measures is 10.4 Gt CO₂e, representing 40.1% of the consumption-based emissions of the 116 countries analysed in this study in 2017. To enhance clarity, we have revised the explanation in the "Data Sources and Processing" section:

We first linked global MRIO data with detailed household expenditure data. The original global MRIO table is sourced from the Global Trade Analysis Project (GTAP) 11 Database (pre-release version), offering detailed information on interregional and intersectoral transactions of the world economy in 2017. It encompasses 160 economies (141 countries and 19 aggregated regions) with 65 economic sectors each ³. GTAP has only one vector for household final consumption for each country. To better capture different expenditure patterns within each country, we used a modified household expenditure database derived from the World Bank's Global Consumption Database (WBGCD) ⁴⁻⁶. The modified WBGCD covers detailed consumption information across 33 categories of consumption items and 201 expenditure levels (i.e., expenditure groups) spanning 116 countries. This accounts for 87.4% of the global population and 80.3% of the global GDP, with substantial representation from developing countries. These datasets are the most detailed available to date. We then developed a bridging matrix to link these 33 consumption categories to the 65 sectors in the GTAP MRIO table, following our previous studies ⁷⁻⁹. Using this matrix, we calculated the consumption shares for each sector by expenditure groups for the 65 sectors in the GTAP. We focused on using the expenditure shares from the WBGCD instead of absolute expenditure values provided by the global consumption database. This approach yielded household final demand data that aligns with the GTAP sectoral classification and maintains consistency across various consumption segments while preserving the detailed differentiation among expenditure groups offered by the WBGCD.

Reviewer 3:

Thank you for your thorough and ambitious response to my previous comments. The additional analyses you have conducted, especially the case studies based on physical data for Switzerland and China, reinforce your findings and rebut my previous concerns. Your willingness to delve deeper is greatly appreciated.

Before I can fully recommend your manuscript for publication, I would like to see a methodological Clarification on Physical Data: It is not entirely clear how you integrated the physical consumption data from your case studies into the emissions calculations. The reference provided does not explain the use of physical data. Providing a more detailed explanation in the supplementary would help readers understand, verify, and potentially replicate your approach here. I realize this is some additional work but ensuring traceability in these calculations is important for building on your work in the future.

RE: thank you for the valuable suggestion. We have added a more detailed data processing explanation in the Supplementary Methods:

Switzerland case study

Switzerland provides highly detailed household expenditure data via its Household Budget Survey (HBS)¹⁰. The HBS dataset for 2015-2017 includes information on both physical consumption and monetary expenditure data for 105 food products across five income groups (monthly income ranging from 0 to 12,856 Swiss francs)¹¹.

To calculate food-related carbon footprints for each income group using both physical and monetary units, we linked the HBS data with Switzerland's household final demand data from the GTAP MRIO dataset. The process, using physical consumption data, involved the following steps:

- 1) Generating Consumption Structure Data: we used physical consumption data from the HBS (e.g., rice consumption in kilograms across five income groups), to calculate proportionate shares for each food item across five income groups. This standardization provides a clear picture of how national consumption for each food product is distributed among different income groups.*
- 2) Mapping HBS Products to GTAP Sectors: we developed a bridging matrix (Supplementary Table 4) to link these 105 food products in the HBS to the 18 food-related sectors within the GTAP dataset, adhering to the sector definitions from the Swiss Federal Statistical Office and existing studies^{10,11}. This allowed us to map the standardized consumption structure data from Step 1 into the 18 GTAP sectors.*
- 3) Disaggregating GTAP household Final Demand: Using the mapped consumption structure data generated in Step 2), we disaggregated the GTAP MRIO household final demand for Switzerland across each of the food-related sectors into the five income groups.*
- 4) Carbon Footprint Calculation: The disaggregated final demand data were used to calculate the carbon footprints for each income group following the same approach outlined in the main text (Baseline household carbon footprints).*

China case study

We grouped China's household survey data into deciles based on their expenditure levels. These subgroup consumption patterns were then bridged with China's household final demand data from the GTAP MRIO data to calculate carbon footprints for each expenditure group in physical and monetary units. The data processing and calculation followed a similar approach as we used for Switzerland. For products lacking

physical data, such as school accommodation fees, monetary expenditures were used as proxies for physical consumption, as is commonly used in the literature^{2,10}.

I also have a comment on Figure 6: You note that the differences in emission reduction potential between “carbon-exceeding” households and average households are large but the figure might give the reader the impression that it is in fact rather small (looks like the mean difference is 10% larger emission reductions from the carbon-exceeding" group. It might be worth acknowledging that in wealthier countries, the “carbon-exceeding” group constitutes a substantial portion of the randomly selected cohort. This could explain why the observed differences appear modest and merit explicit mention in the text. Alternatively, think of how this figure could be changed to better reflect your conclusions. This is just a suggestion, do as you please.

RE: Thank you for your suggestion. We have expanded the description for Figure 6 as follows:

In North American and European nations, the observed deviations remain relatively modest despite a high potential for emission reductions. This can be primarily attributed to the substantial presence of carbon-exceeding households within the randomly selected average consumers in these affluent nations.

If you address the above point, I would be ready to recommend it for publication. You have already made significant efforts to improve the work, and these clarifications will strengthen its overall contribution to the literature.

RE: We deeply appreciate the constructive feedback you provided, which has significantly improved our paper and offered new perspectives for future research.

References

1. Hardadi, G., Buchholz, A. & Pauliuk, S. Implications of the distribution of German household environmental footprints across income groups for integrating environmental and social policy design. *Journal of Industrial Ecology* **25**, 95–113 (2021).
2. Zhang, Y., Wang, F. & Zhang, B. The impacts of household structure transitions on household carbon emissions in China. *Ecological Economics* **206**, 107734 (2023).
3. Aguiar, A., Chepeliev, M., Corong, E. & Mensbrugge, D. van der. The Global Trade Analysis Project (GTAP) Data Base: Version 11. *Journal of Global Economic Analysis* **7**, (2022).
4. The World Bank. Global Consumption Database. <https://datatopics.worldbank.org/consumption/> (2024).
5. Li, Y. *et al.* Reducing climate change impacts from the global food system through diet shifts. *Nat. Clim. Chang.* 1–11 (2024) doi:10.1038/s41558-024-02084-1.
6. Tian, P. *et al.* Keeping the global consumption within the planetary boundaries. *Nature* **635**, 625–630 (2024).
7. Hubacek, K., Baiocchi, G., Feng, K. & Patwardhan, A. Poverty eradication in a carbon constrained world. *Nat Commun* **8**, 912 (2017).
8. Guan, Y. *et al.* Burden of the global energy price crisis on households. *Nat Energy* **8**, 304–316 (2023).
9. Bruckner, B., Hubacek, K., Shan, Y., Zhong, H. & Feng, K. Impacts of poverty alleviation on national and global carbon emissions. *Nat Sustain* **5**, 311–320 (2022).
10. Girod, B. & De Haan, P. More or Better? A Model for Changes in Household Greenhouse Gas Emissions due to Higher Income. *Journal of Industrial Ecology* **14**, 31–49 (2010).
11. Switzerland Federal Statistical Office. Switzerland Household Budget Survey, 2015-2017. <https://www.bfs.admin.ch/bfs/en/home/statistiken/wirtschaftliche-soziale-situation-bevoelkerung/einkommen-verbrauch-vermoegen/haushaltsbudget.html> (2024).

Response letter to Reviewer 3' comment

I'm still unclear whether there is a Chinese HBS survey that collects data in physical units. If so, please provide a specific reference. The current citations aren't sufficient. If the physical data comes only from trade data (e.g., imports/exports in GTAP), it wouldn't address heterogeneity among different household groups.

RE: We used the China Household Survey (CHS) data collected by the National Bureau of Statistics of China (NBS), formerly known as the Urban/Rural Household Income and Expenditure Survey (UHIES/RHIES) before 2012, and integrated into a unified urban-rural survey after 2013¹. Previous studies have primarily focused on pre-2012 UHIES dataset^{2,3}.

As we previously stated in the Supplementary Methods, the 2014 CHS dataset includes both physical and monetary units for 59 products. While the dataset is not publicly available, the NBS confirms that it contains these details, stating: "Household consumption expenditure refers to all expenditures by households, including both monetary and physical consumption." Additionally, the NBS publishes the China Yearbook of Household Survey annually, which provides national-level aggregate data, including both monetary and physical consumption information for key products (e.g., food and durable goods).

In response to this comment, we have added further data descriptions and citations:

We utilized the 2014 China Household Survey (CHS), which was collected by the National Bureau of Statistics of China using a stratified multistage random sampling approach¹. Formerly known as China's Urban/Rural Household Income and Expenditure Survey (UHIES/RHIES), it was integrated into a unified urban-rural survey after 2013¹. The 2014 CHS dataset used in this study includes over 13,000 households across four provinces (Liaoning, Sichuan, Guangdong, and Shanghai). The dataset provides detailed household-level data on demographics (e.g., size, income, dwelling characteristics) and expenditures for 145 products, 59 of which include physical consumption data (e.g., food, electricity, and private cars). A number of studies have recently used the household consumption survey data from this dataset^{2,3}.

References

1. National Bureau of Statistics. Information on China Household Survey. *Household Survey* https://www.stats.gov.cn/hd/cjwtd/202302/t20230207_1902268.html (2023).
2. Zha, D., Su, X. & Al-Samhi, M. M. M. Will rebound behaviour diminish the decarbonization potential of carbon generalized system of preferences in China? *Sustainable Production and Consumption* **47**, 474–484 (2024).
3. Zhang, Y., Wang, F. & Zhang, B. The impacts of household structure transitions on household carbon emissions in China. *Ecological Economics* **206**, 107734 (2023).